# Base Models Know How to Reason, Thinking Models Learn When

**Constantin Venhoff** [* 1 2]  **Iván Arcuschin** [* 3]  **Philip Torr** [1]  **Arthur Conmy**  **Neel Nanda**

## Abstract

What do *thinking* language models learn during training that their base models lack? We first present an unsupervised method that discovers a model's reasoning behaviors by training small Sparse Autoencoders on sentence-level activations of reasoning traces, yielding interpretable reasoning taxonomies. Building on this, we introduce *constructive model diffing*, which aims to reconstruct the base-to-fine-tuned difference from interpretable components: *reasoning mechanisms* (category vectors that can induce a reasoning behavior in the base model) and *reasoning heuristics* (a classifier determining when a mechanism should fire). Across nine base/thinking pairs (four RL-trained, four SFT-distilled, one mixed), two independent findings agree: category vectors in the base model converge to far lower loss for taxonomies derived from purely RL-trained models, and hybrid models recover roughly 76% of the RL base-to-thinking gap but only 11% of the SFT gap. This indicates RL primarily teaches heuristics for orchestrating pre-existing base mechanisms, whereas SFT-distillation installs new ones, offering a new lens on what training paradigms teach, with implications for efficient reasoning-model development.

## 1. Introduction

Large Language Models (LLMs) have recently demonstrated remarkable capabilities in reasoning tasks when given additional inference time to think through problems step-by-step. *Thinking models*, such as OpenAI's o3 (OpenAI, 2025), DeepSeek's R1 (Guo et al., 2025), QwQ-32B (Qwen Team, 2024), and Open-Reasoner-Zero (Hu et al., 2025), significantly outperform their base counterparts on

challenging reasoning benchmarks (Chollet, 2024). However, a fundamental question remains: *What exactly do thinking models learn that their base counterparts lack?*

Prior work has suggested several hypotheses: thinking models may acquire entirely new reasoning capabilities (Gandhi et al., 2025), learn to structure reasoning more effectively (Marjanović et al., 2025), repurpose pre-existing representations (Ward et al., 2025a), or simply benefit from additional inference-time computation (Zhao et al., 2025). In this paper, we introduce a methodology to directly analyze what thinking models learn, and find that the answer depends critically on *how* they were trained.

We make the following contributions:

1. We develop an **unsupervised methodology for discovering reasoning behaviors** in thinking models using Sparse Autoencoders (SAEs), producing interpretable taxonomies of the cognitive operations these models employ (Section 2).
2. We introduce **constructive model diffing**, a framework for understanding what fine-tuned models learn by explicitly constructing the base-to-fine-tuned difference and measuring how well this construction recovers the fine-tuned model's performance (Section 3).
3. Applying this framework, we find **empirical evidence that RL-trained and SFT-distilled thinking models learn fundamentally different things**, observing striking performance recovery differences (roughly 76% vs. 11% on average) between the two training paradigms (Section 3.7).

Our constructive model diffing approach decomposes the base-to-thinking model difference into two components: (1) *reasoning mechanisms*, represented as category vectors that induce specific behaviors in the base model, and (2) *reasoning heuristics*, a classifier extracted from the thinking model that determines when each mechanism should fire. By combining base model generation with thinking model heuristics and base model category vectors, we can explicitly reconstruct the diff and measure its fidelity.

Evaluating nine configurations (0.5B to 32B parameters), the RL-trained models (Open-Reasoner-Zero) achieve roughly 76% average recovery across GSM8K, MATH500, and a held-out Hendrycks-MATH set while steering only

---
[*]Equal contribution   [1]University of Oxford, UK [2]MATS [3]Poseidon Research.   Correspondence to:   Constantin Venhoff <constantin@robots.ox.ac.uk>,   Iván   Arcuschin <ivan@poseidonresearch.com>.

*Proceedings of the 43$^{rd}$ International Conference on Machine Learning*, Seoul, South Korea. PMLR 306, 2026. Copyright 2026 by the author(s).

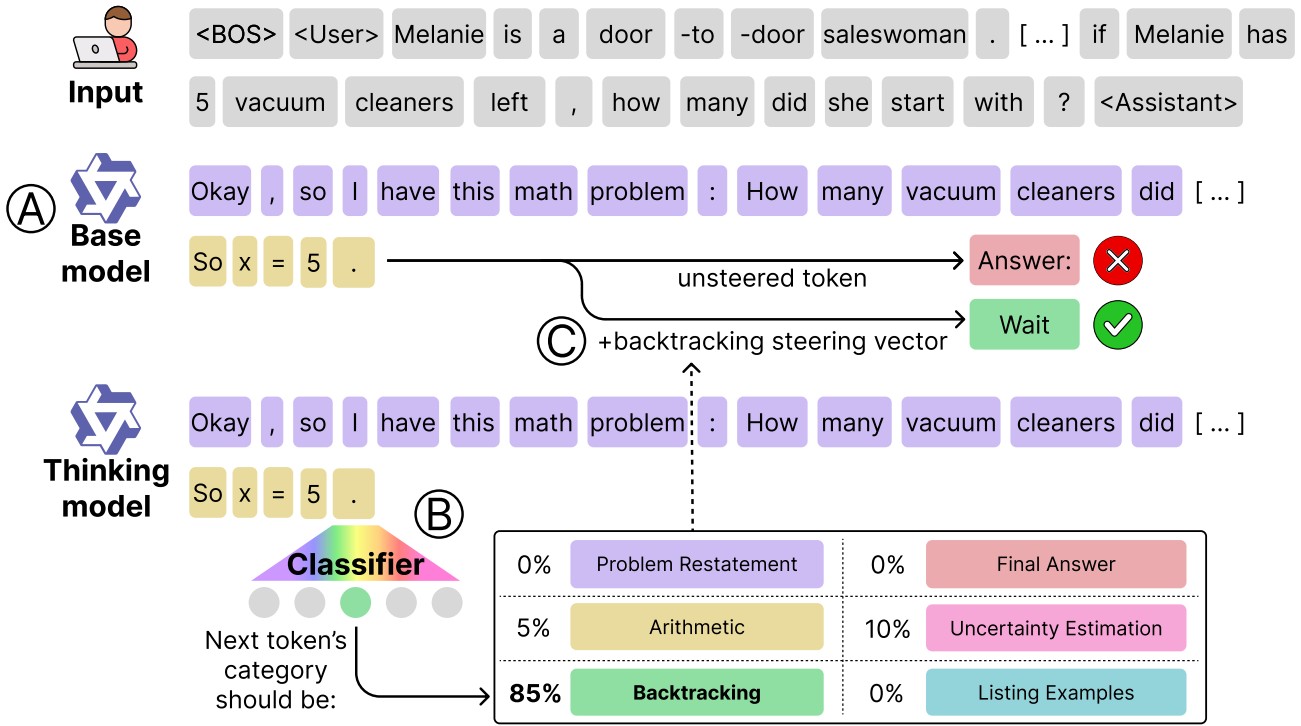

*Figure 1.* **Constructive Model Diffing for Thinking LLMs.** We decompose the difference between base and thinking models into two interpretable components. **(A)** The **base model** generates tokens, providing the underlying reasoning capabilities. **(B)** A **reasoning heuristic**, extracted from the thinking model via SAE activations, determines *when* to deploy each reasoning mechanism. **(C)** **Category vectors**, optimized in the base model, induce the corresponding reasoning behavior when triggered. If this explicit construction recovers the thinking model's performance, it provides evidence that the diff is well-characterized by our decomposition: the thinking model learned sophisticated heuristics over pre-existing base model mechanisms.

a small fraction of tokens (around 5–12%), whereas SFT-distilled models (DeepSeek-R1-Distill) recover far less on average, with meaningful recovery emerging only for the larger base models. Since both model types reach comparable benchmark performance and produce similar-quality taxonomies, the gap must stem from the category vectors themselves. This implies RL teaches primarily heuristics for the base-model's pre-existing mechanisms, while SFT-distillation modifies the mechanisms by training on teacher demonstrations that may use different reasoning behaviors than the base model.

To ease reproducibility and further research, we publish our codebase and results in a public GitHub repository[1].

## 2. Taxonomy of Reasoning Mechanisms

Recent work on thinking models has primarily relied on manual inspection of the model's reasoning traces to identify the underlying mechanisms it uses to perform reasoning (see Section 4). While insightful, such approaches are inherently subjective and may overlook subtle or distributed reasoning patterns. To support our main analysis, we develop an unsupervised, bottom-up methodology to discover human-interpretable reasoning mechanisms in thinking models. Our goal is to construct a taxonomy of reasoning mechanisms that is:

1. **Interpretable:** Each reasoning mechanism should be understandable by humans, with a clear description of its cognitive function and role in the reasoning process.
2. **Complete:** The taxonomy should cover the full range of types of reasoning steps the model can use, ensuring no significant patterns of reasoning are overlooked in our analysis.
3. **Independent:** The categories should correspond to distinct cognitive functions with minimal overlap between different reasoning processes.

### 2.1. Unsupervised Clustering via Sparse Autoencoders

Unsupervised methods are useful for building a taxonomy of reasoning mechanisms, as they let us discover reasoning patterns without imposing pre-existing assumptions about how models reason. Clustering algorithms are particularly well-suited, identifying natural groupings in high-dimensional

---

[1]https://github.com/cvenhoff/thinking-llms-interp

activation spaces that correspond to distinct reasoning functions.

Sparse Autoencoders (SAEs) (Olshausen & Field, 1997; Lee et al., 2006) have become a popular tool for decomposing Large Language Model (LLM) activations into interpretable features (Cunningham et al., 2024; Bricken et al., 2023; Templeton et al., 2024). We use Top-K SAEs (Makhzani & Frey, 2014; Gao et al., 2025), which retain only the $K$ largest-magnitude latent components, with a deliberately small dictionary and low $k$ (see Appendix A for details). This configuration directly mirrors our hypotheses about reasoning: the dictionary size sets the number of distinct reasoning mechanisms we expect a model to use, while $k$ bounds how many can be simultaneously active within a single sentence.

With such a restricted dictionary, the SAE is forced to learn the subspace components that best explain the variance of our sentence activations, effectively making it a subspace clustering method. Concretely, we restrict the latent dimension to the range $[5, 50]$, far smaller than the input dimension (e.g., 1,536 for Qwen2.5-1.5B) and unlike the much larger dictionaries typical in Mechanistic Interpretability (Templeton et al., 2024; Gao et al., 2025), so that the discovered features capture the most fundamental axes of reasoning variation rather than incidental linguistic detail.

### 2.2. SAE Training and Evaluation

We now describe how we train SAEs on reasoning data and how we turn the resulting clusters into a scored human-interpretable taxonomy.

**SAE taxonomy training.** We train our Top-K SAEs on sentence-level activations extracted from reasoning rollouts that each thinking model generates. We operate at the sentence level because sentences offer an intermediate abstraction that avoids the excessive granularity of token-level analysis while retaining more precision than paragraph-level approaches, and prior work shows that individual sentences in reasoning traces perform distinct functions (Bogdan et al., 2025; Venhoff et al., 2025; Nye et al., 2021). We therefore average token activations within each sentence, under the assumption that each sentence is primarily characterized by one to three reasoning categories. Full data, layer choices, and hyperparameters are given in Appendix A.

**Taxonomy interpretation.** To derive human-understandable categories from our SAE representations, we use an LLM-based interpretability approach. For each cluster, we collect 100 top exemplar sentences that most strongly activate the feature and 100 random sentences from the same cluster, then prompt an LLM to identify the precise cognitive function these sentences serve in the

reasoning process. Crucially, the categories themselves are discovered by the SAE rather than supplied by the LLM, so this labeling step does not reintroduce top-down assumptions about which mechanisms should exist.

This process generates our list of interpretable reasoning categories with their titles and descriptions, which forms the foundation for our subsequent evaluation metrics. See Appendix B.1 for the complete cluster generation prompt and more details.

### 2.3. Taxonomy Evaluation Metrics

Since we do not know *a priori* how many clusters or which layer best captures the reasoning mechanisms, we need a robust way to evaluate and compare SAE configurations. We score each candidate taxonomy on the sentences collected from each thinking model's reasoning traces on MMLU-Pro prompts, using the average of three components that map directly onto our objectives: *completeness*, *consistency*, and *independence*.

**Consistency.** We measure the consistency of our categories by evaluating how well an LLM can classify individual sentences from within and outside each category using the generated titles and descriptions. For a given cluster, we take the average F1 score across all categories as our overall consistency score.

**Completeness.** We measure the completeness of our categories by evaluating the confidence that an LLM has in classifying individual sentences into their assigned categories.

**Independence.** We measure the independence of our categories by asking an LLM to evaluate how semantically similar all pairs of categories are in a cluster. This information is then used to calculate the fraction of pairs with similarity below a threshold ($0.5$), equivalently, those with orthogonality above $0.5$, which we consider functionally distinct.

For all these metrics, the higher the value, the better the taxonomy. More details on the prompts and specific implementation are provided in Appendix B.

### 2.4. Taxonomy Results

To evaluate our approach to building interpretable taxonomies, we analyze nine thinking models: four SFT-distilled variants (DeepSeek-R1-Distill-`Llama-8B`, `Qwen-1.5B`, `Qwen-14B`, `Qwen-32B`), four RL-trained models (Open-Reasoner-Zero: `ORZ-0.5B`, `ORZ-1.5B`, `ORZ-7B`, `ORZ-32B`) (Hu et al., 2025), and one mixed model (`QwQ-32B`, trained first with SFT on reasoning demonstrations and then with RL (Qwen Team, 2024)).

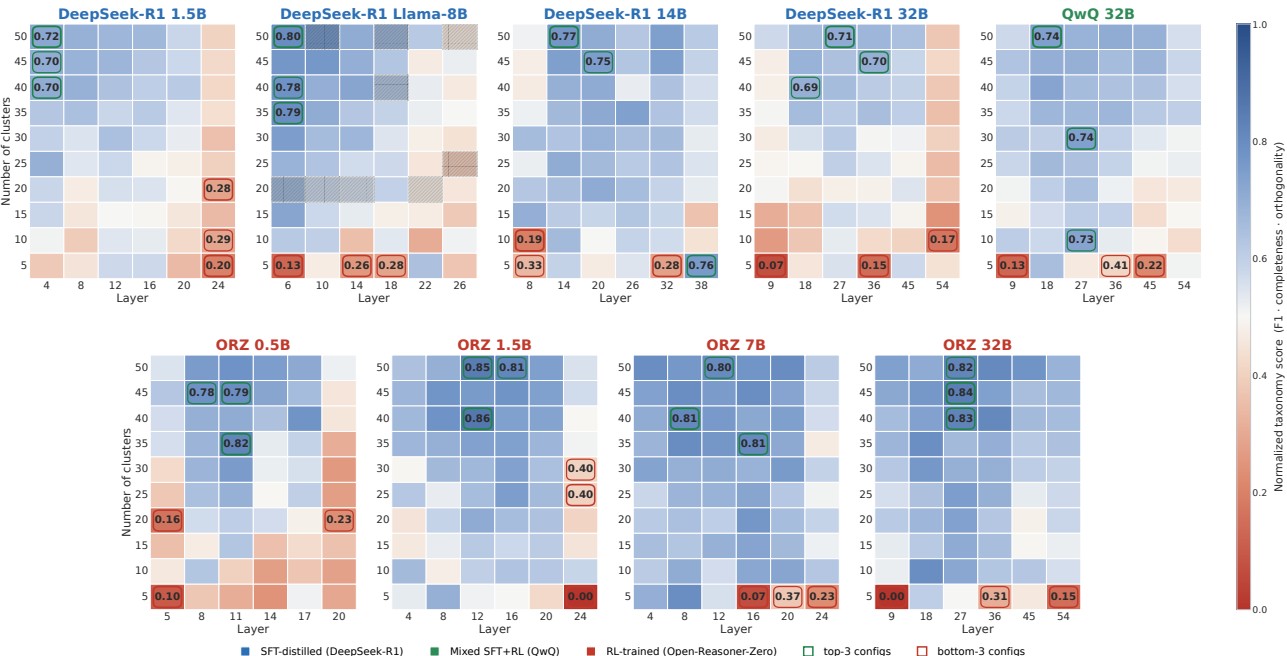

*Figure 2.* **Grid search over SAE taxonomies.** Combined score (average of completeness, independence, and consistency) across layers (x-axis) and cluster sizes (y-axis, 5–50), for DeepSeek-R1 SFT-distilled variants and QwQ-32B (top) and Open-Reasoner-Zero models (bottom). The three best configurations per model are outlined in green, the three worst in red. While scores rise with cluster size, an "elbow" at 10–20 categories indicates reasoning mechanisms are already well represented at that scale in aggregate; individual models' elbows can sit lower, so the per-model taxonomies we ultimately select (Appendix G) use dictionary sizes in the 5–15 range. The colorbar's "F1 · completeness · orthogonality" label denotes this same combined score, where "F1" and "orthogonality" are the consistency and independence metrics defined in Section 2.3; the three components are averaged, not multiplied.

We performed an extensive grid search across these nine models, using 6 distributed layers and cluster sizes (ranging from 5 to 50 categories with increments of 5) to identify the optimal taxonomy configuration. For comparison across configurations, we apply min-max normalization within each model. The results are shown in Figure 2, and we provide the complete taxonomies for our selected SAE configurations in Appendix G. To confirm that these results are not an artifact of using LLMs as graders, we validate both the grading judges and the discovered category labels against human annotators in Appendix C, finding strong agreement (e.g. 14/15 ORZ-7B categories match a human-written label).

### 2.5. Stability of Discovered Features

SAEs trained on the same data can learn different features depending on the random seed (Paulo & Belrose, 2026). To test the stability of our taxonomies we replicate the protocol of Paulo & Belrose (2026) on an ORZ-7B SAE (layer 16, dictionary size 15): we retrain it five times with different seeds, match features one-to-one between every pair of runs via the Hungarian algorithm on cosine similarity, and count pairs with cosine $> 0.7$ as "matched." Despite genuinely different training dynamics across seeds (early stopping be-

tween epochs 17 and 35), the SAEs are highly consistent, with a 93.3% average feature-matching rate and mean pairwise cosine of 0.886 across all ten comparisons (Table 3). This far exceeds the $\sim 30\%$ matching rate Paulo & Belrose (2026) report for large SAEs, and is consistent with their observation that smaller models and smaller dictionaries share more features. Our low-dimensional, sentence-level SAEs thus operate well away from the high-dimensional regime they identify as unstable, indicating that the discovered taxonomy is driven primarily by the underlying activation geometry rather than by seed randomness.

## 3. Constructive Model Diffing

Having established an unsupervised taxonomy of reasoning mechanisms (Section 2), we now introduce *constructive model diffing*: a framework for understanding what fine-tuned models learn by explicitly constructing the base-to-fine-tuned difference.

### 3.1. Motivation

Traditional approaches to model diffing typically apply post-hoc analysis to identify which representations are shared versus specific between model versions. For instance, cross-

coders (Lindsey et al., 2024) have been used to find features specific to chat fine-tuning by comparing base and instruction-tuned models (Minder et al., 2025). While effective for discovering fine-grained feature-level differences, such methods face a conceptual limitation: they assume the diff can be explained at the level of linear features.

For understanding thinking models, we hypothesize that the diff operates at a higher level of abstraction. Following Marjanović et al. (2025), we define a *reasoning mechanism* as an individual cognitive-like operation (e.g., verifying an intermediate result, backtracking, setting a subgoal) that serves as a compositional building block of the reasoning process. Our hypothesis is that thinking models may learn a sophisticated *heuristic* for when to deploy these mechanisms, rather than learning the mechanisms themselves. Such a heuristic, which coordinates multiple mechanisms across a reasoning trace, is difficult to decompose into individual linear features.

Constructive model diffing addresses this by taking a different approach: rather than decomposing the diff post-hoc, we *construct* it explicitly from interpretable components and measure how well this construction recovers the fine-tuned model's performance. High recovery indicates that our decomposition captures the essential difference; low recovery suggests the fine-tuned model learned something our construction cannot represent.

### 3.2. Framework

We decompose the base-to-thinking model difference into two components:

- **Reasoning mechanisms:** Category vectors optimized in the *base model* that, when applied, induce specific reasoning behaviors. These represent the "what" of reasoning.
- **Reasoning heuristics:** A classifier derived from the *thinking model* that determines when each mechanism should be activated. This represents the "when" of reasoning.

Our *hybrid model* combines these components: the base model generates tokens, while the thinking model's heuristic (via SAE activations) triggers category vectors at appropriate moments. If this construction recovers thinking model performance, it provides evidence that the thinking model primarily learned the heuristic over pre-existing base model capabilities.

### 3.3. Training Category Vectors

We leverage the taxonomy from Section 2 to construct category vectors in the base model. Category vectors are steering vectors (Turner et al., 2023; Arditi et al., 2024; Zou et al., 2023; Rimsky et al., 2024), i.e., directions in activation space that, when added to intermediate activations, induce target behaviors.

Since SAEs identify variance-explaining rather than causally important directions, we optimize category vectors to find causal directions corresponding to each SAE-discovered mechanism:

1. Generate thinking-model rollouts on a diverse training mix and teacher-force them through the base model, identifying token positions where the two models disagree.
2. At each disagreement position, classify the reasoning category by running the thinking model's hidden state through the SAE encoder and taking the top activating category as the label.
3. Jointly optimize a per-category vector and a small MLP that predicts a steering coefficient, minimizing cross-entropy on the thinking model's next token at disagreement positions.

Based on our taxonomy grid search (Figure 2), we select layer and cluster sizes at the performance elbow. The complete taxonomies used for all nine model pairs are listed in Appendix G. We optimize category vectors at approximately 37% of model depth, which prior work suggests is most causal for behavior modification (Venhoff et al., 2025). To control for run-to-run variance, we train three independent sets of category vectors per model and, for each set, build a hybrid model and measure its gap recovered on a "holdout mix" (a gold-answer subset of the training-data validation split, never used for gradient updates). We select the set with the highest holdout-mix gap recovered, and use only that set for all evaluation benchmarks. Full training details are in Appendices E.1 and E.2.

### 3.4. Training Results

Figure 3 shows the category-balanced cross-entropy on three held-out sets during training. Across all splits, the RL-trained ORZ models converge to substantially lower cross-entropy than the SFT-distilled R1 models, or the mixed QwQ model, indicating that category vectors trained on ORZ pairs find representations that more effectively nudge the base model toward the SAE-discovered reasoning categories.

### 3.5. Extracting the Reasoning Heuristics

The second component of our construction is the heuristic: the decision of *when* to activate each reasoning mechanism. We extract this in two steps. First, we check whether the base- and the thinking model disagree in their next token prediction. If not we generate the jointly predicted tokens. If they disagree, then we obtain the SAE activations at that token position and select the highest activating category as the category to steer the base model with.

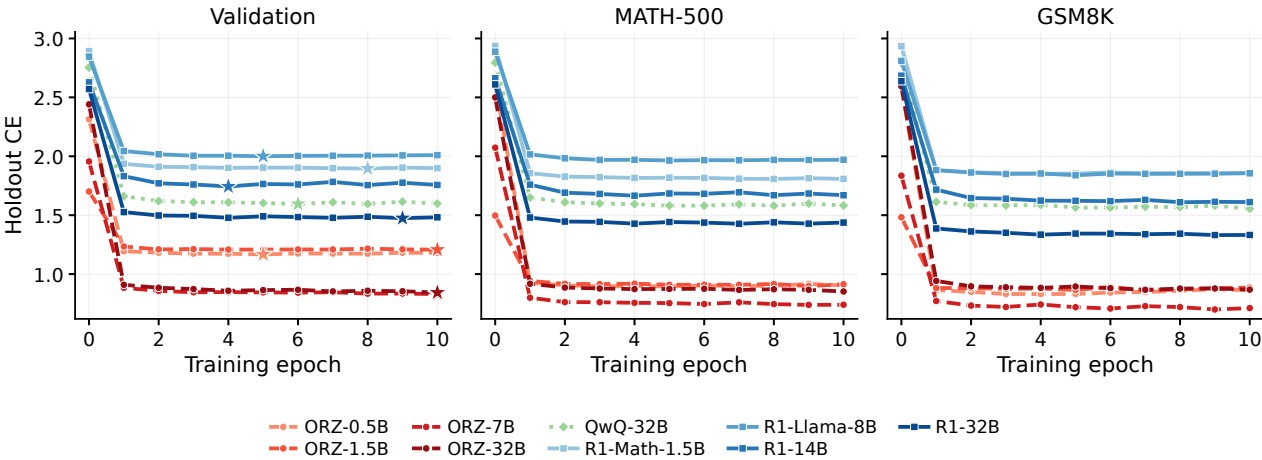

*Figure 3.* **Category vector training: holdout loss vs. epoch.** RL-trained ORZ models (warm colours) converge to lower cross-entropy than SFT-distilled R1 models (cool colours) across all three held-out sets, indicating that ORZ-derived category vectors find more causally effective directions in the base model. The mixed-training QwQ model falls in between. Stars mark the best epoch per model on the validation split.

### 3.6. The Hybrid Model

Our hybrid model explicitly reconstructs the base-to-thinking model diff:

- The **base model** generates tokens and provides the underlying reasoning capabilities.
- The **thinking model's heuristic** (via SAE activations and disagreement gating) decides which reasoning behaviour to steer towards.
- The **category vectors** induce the appropriate reasoning mechanism when triggered.

During generation, we compute SAE activations at each disagreement position and apply the selected category vector. The steering strength is provided by the jointly trained coefficient MLP. For fair comparison, both base-only and hybrid models use identical prompts (see Appendix E.3).

This construction is interpretable by design: with only 5-15 distinct category vectors applied sparsely, the hybrid cannot succeed by memorizing outputs. If it recovers thinking model performance, this demonstrates that the diff is well-characterized by our mechanism + heuristic decomposition.

### 3.7. Evaluation Results: RL vs. SFT-Distillation

We evaluate constructive model diffing across nine base-/thinking model pairs (four RL-trained, four SFT-distilled, and one mixed) on GSM8K (Cobbe et al., 2021), MATH500 (Hendrycks et al., 2021), and a held-out Hendrycks-MATH subset disjoint from training and MATH500. Table 1 reveals a striking difference: RL-trained models (Open-Reasoner-Zero (Hu et al., 2025)) recover roughly 76% of the base-to-thinking gap on average across the three benchmarks. The

SFT-distilled models (DeepSeek-R1-Distill) recover far less: the two smallest pairs recover essentially nothing, while recovery grows with base-model size, reaching ∼15–20% for the 14B and 32B distilled models. QwQ-32B (Qwen Team, 2024), which combines SFT on reasoning demonstrations with subsequent RL, recovers ∼20%. In both size-matched comparisons (1.5B and 32B) the RL-trained pair recovers substantially more; in particular, at 32B the RL-trained Open-Reasoner-Zero-32B pair clearly outperforms both the same-size R1-Distill-32B and QwQ-32B pairs.

**Why the difference?** Our constructive diff has two components: (1) the SAE-based heuristic from the thinking model, and (2) category vectors that induce the thinking LLM's reasoning behaviors in the respective base model. Critically, both SFT-distilled and RL thinking models achieve similar benchmark performance (at comparable sizes), and our taxonomies achieve similar quality across model types (Figure 2). Since these factors are comparable, the recovery gap must stem from the category vectors: they effectively induce reasoning for RL models but not for SFT-distilled models.

**Interpretation.** This implies that RL models use reasoning behaviors that their base model counterpart already knows, and therefore primarily learn sophisticated heuristics for orchestrating those pre-existing base model behaviors, which is why our decomposition captures their behavior well. Distilled and mixed models (DeepSeek-R1-Distill, and QwQ-32B which adds RL on top of SFT), in contrast, learn or substantially refine their reasoning mechanisms during training, making base model steering insufficient to fully replicate their behaviors. The clear size trend among

*Table 1.* **Constructive model diffing results on GSM8K, MATH500, and a held-out Hendrycks-MATH subset.** We compare each base, thinking, and hybrid model triplet. Base and hybrid models decode greedily; thinking models sample 3 rollouts at temperature 0.6. All use a 2048-token budget. Accuracy is graded by Claude Sonnet 4.6 (Anthropic, 2025), which agrees closely with human grading ($\kappa = 0.88$; Appendix C); we run 3 judge repetitions per rollout and report mean $\pm$ std over all rollout-judge pairs. The Hendrycks-MATH set is a 1,000-question subset of Hendrycks et al. (2021), disjoint from both the training mix and MATH500. Rec.% is the fraction of the base-to-thinking gap that the hybrid recovers

| | | GSM8K | | | | MATH500 | | | | Hendrycks-MATH | | | |
|---|---|---|---|---|---|---|---|---|---|---|---|---|---|
| Base Model | Thinking Model | Base | Hybrid | Think | Rec.% | Base | Hybrid | Think | Rec.% | Base | Hybrid | Think | Rec.% |
| *SFT-Distilled Thinking Models* | | | | | | | | | | | | | |
| Qwen2.5-Math-1.5B | R1-Distill-1.5B | $74.0_{\pm0.1}$ | $73.5_{\pm0.1}$ | $90.8_{\pm0.2}$ | ↓-3.0 | $66.5_{\pm0.2}$ | $68.1_{\pm0.2}$ | $93.4_{\pm1.0}$ | ↑5.7 | $71.7_{\pm0.1}$ | $69.8_{\pm0.2}$ | $94.1_{\pm0.7}$ | ↓-8.5 |
| Llama-3.1-8B | R1-Distill-8B | $45.6_{\pm0.2}$ | $44.0_{\pm0.3}$ | $94.3_{\pm0.2}$ | ↓-3.4 | $32.3_{\pm0.1}$ | $33.1_{\pm0.1}$ | $94.0_{\pm0.1}$ | ↑1.2 | $30.2_{\pm0.1}$ | $26.6_{\pm0.2}$ | $95.8_{\pm0.2}$ | ↓-5.5 |
| Qwen2.5-14B | R1-Distill-14B | $77.3_{\pm0.3}$ | $81.7_{\pm0.1}$ | $97.2_{\pm0.4}$ | ↑22.2 | $63.4_{\pm0.2}$ | $69.5_{\pm0.1}$ | $97.0_{\pm0.2}$ | ↑18.3 | $67.7_{\pm0.1}$ | $72.2_{\pm0.1}$ | $97.1_{\pm0.2}$ | ↑15.4 |
| Qwen2.5-32B | R1-Distill-32B | $69.4_{\pm0.1}$ | $78.4_{\pm0.1}$ | $97.6_{\pm0.2}$ | ↑31.8 | $61.0_{\pm0.2}$ | $66.4_{\pm0.3}$ | $97.5_{\pm0.2}$ | ↑14.8 | $65.7_{\pm0.3}$ | $69.7_{\pm0.0}$ | $97.5_{\pm0.3}$ | ↑12.5 |
| *RL-Trained Thinking Models* | | | | | | | | | | | | | |
| Qwen2.5-0.5B | ORZ-0.5B | $25.0_{\pm0.2}$ | $43.2_{\pm0.0}$ | $47.0_{\pm0.5}$ | ↑83.0 | $27.3_{\pm0.1}$ | $36.2_{\pm0.3}$ | $36.6_{\pm0.5}$ | ↑95.3 | $28.8_{\pm0.0}$ | $39.2_{\pm0.1}$ | $38.5_{\pm0.5}$ | ↑107.4 |
| Qwen2.5-1.5B | ORZ-1.5B | $51.9_{\pm0.1}$ | $68.1_{\pm0.1}$ | $74.7_{\pm0.8}$ | ↑71.3 | $39.1_{\pm0.1}$ | $55.2_{\pm0.2}$ | $58.6_{\pm1.0}$ | ↑82.7 | $39.0_{\pm0.2}$ | $60.3_{\pm0.1}$ | $63.1_{\pm0.4}$ | ↑88.3 |
| Qwen2.5-7B | ORZ-7B | $71.5_{\pm0.2}$ | $90.7_{\pm0.0}$ | $93.4_{\pm0.2}$ | ↑87.8 | $63.9_{\pm0.1}$ | $75.7_{\pm0.1}$ | $87.6_{\pm0.4}$ | ↑49.7 | $66.9_{\pm0.1}$ | $77.7_{\pm0.2}$ | $89.0_{\pm0.3}$ | ↑49.1 |
| Qwen2.5-32B | ORZ-32B | $69.5_{\pm0.1}$ | $95.5_{\pm0.0}$ | $97.2_{\pm0.3}$ | ↑93.6 | $61.1_{\pm0.2}$ | $81.8_{\pm0.0}$ | $95.8_{\pm1.1}$ | ↑59.8 | $65.7_{\pm0.1}$ | $80.6_{\pm0.0}$ | $95.9_{\pm0.4}$ | ↑49.2 |
| *Mixed (SFT + RL)* | | | | | | | | | | | | | |
| Qwen2.5-32B | QwQ-32B | $69.7_{\pm0.1}$ | $77.9_{\pm0.1}$ | $97.9_{\pm0.2}$ | ↑29.1 | $60.9_{\pm0.1}$ | $66.3_{\pm0.1}$ | $98.3_{\pm0.4}$ | ↑14.3 | $65.6_{\pm0.1}$ | $70.9_{\pm0.1}$ | $98.0_{\pm0.2}$ | ↑16.3 |

*Table 2.* **Steered token fraction.** Average fraction of tokens receiving steering per problem.

| Base Model | Thinking Model | GSM8K | MATH500 | Hendrycks |
|---|---|---|---|---|
| *SFT-Distilled Thinking Models* | | | | |
| Qwen2.5-Math-1.5B | R1-Distill-1.5B | 15.8% | 12.0% | 12.1% |
| Llama-3.1-8B | R1-Distill-8B | 17.6% | 12.5% | 13.4% |
| Qwen2.5-14B | R1-Distill-14B | 15.0% | 12.9% | 13.5% |
| Qwen2.5-32B | R1-Distill-32B | 23.3% | 23.4% | 23.7% |
| *RL-Trained Thinking Models* | | | | |
| Qwen2.5-0.5B | ORZ-0.5B | 7.1% | 5.9% | 5.3% |
| Qwen2.5-1.5B | ORZ-1.5B | 8.8% | 4.9% | 5.0% |
| Qwen2.5-7B | ORZ-7B | 6.9% | 5.9% | 6.0% |
| Qwen2.5-32B | ORZ-32B | 11.5% | 12.2% | 12.2% |
| *Mixed (SFT + RL)* | | | | |
| Qwen2.5-32B | QwQ-32B | 28.6% | 22.7% | 22.0% |

new ones.

**Sparse interventions suffice.** As shown in Table 2, the RL-trained ORZ models – those that recover most of the base–thinking gap – steer only ∼5–12% of tokens per problem, and even the highest-steering pairs stay well below a third of tokens. This sparsity rules out the hypothesis that steering simply biases toward specific outputs: with only a handful of category vectors applied intermittently, there is insufficient information to generate correct answers through token-level manipulation alone. See Figure 5 in Appendix D.

### 3.8. Ablation Studies

To test whether each component of our construction is necessary, we run four negative-control ablations on two RL pairs chosen to span a small and a large model, Open-Reasoner-Zero-1.5B and Open-Reasoner-Zero-32B, reporting *gap recovered* (the fraction of the base-to-thinking accuracy gap closed by the hybrid) averaged over MATH500 and GSM8K (Figure 4). The full pipeline recovers ∼77% of the gap for both models; each ablation removes one ingredient.

- **Random category.** Steering with a randomly chosen category instead of the SAE-selected one drops recovery to 20–28%, showing that *which* mechanism is applied matters.
- **Random vectors.** Replacing the learned category vectors with norm-matched random directions, while keeping the trained MLP *and* the real SAE category selection, drives recovery below zero ($-28\%$ and $-13\%$): with the correct category still chosen at each position, the MLP applies confident coefficients along meaningless directions and actively harms the model, show-

these models is telling: the larger distilled and mixed models (R1-Distill-14B, R1-Distill-32B, and QwQ-32B) recover noticeably more of the gap than the smaller ones, suggesting that larger base models already "know" more of the reasoning mechanisms present in the SFT training data simply by virtue of scale, and can therefore represent more of them as category vectors. Crucially, this trend does not overturn our conclusion: even at 32B, the RL-trained Open-Reasoner-Zero-32B outperforms both R1-Distill-32B and QwQ-32B in category-vector training loss and hybrid recovery, showing that at every scale RL primarily orchestrates pre-existing mechanisms while SFT-based distillation teaches new ones.

Intuitively, SFT-distillation trains a model to imitate a teacher that may employ reasoning mechanisms the base model does not know, forcing the student to modify its mechanisms. RL training optimizes the model's own outputs against a reward signal, encouraging it to leverage what it already knows. The performance boost comes from learning *when* to deploy existing capabilities, not from learning

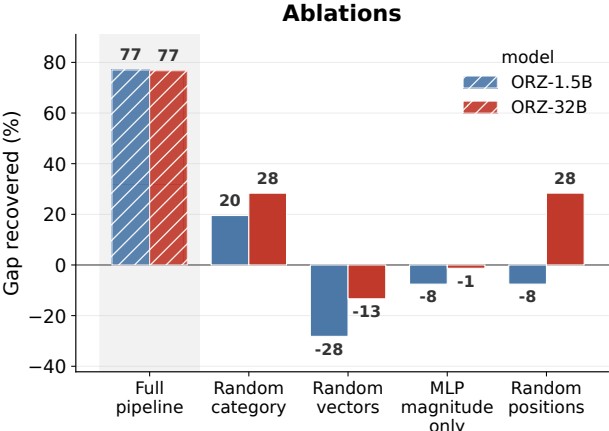

*Figure 4.* **Negative-control ablations.** We run hybrid evaluation on MATH500 and GSM8K for Open-Reasoner-Zero-1.5B and -32B while ablating various hybrid-model components. "Full pipeline" bars are the baseline gap-recovery results; every other bar removes a single component. All ablations sharply reduce recovery, confirming that each component contributes.

ing that the learned directions, not the MLP, carry the causal signal.

- **MLP magnitude only.** We combine both perturbations above, randomizing the directions *and* which category fires, so the trained coefficient MLP is the only intact component and its per-position magnitude prediction is the sole genuine signal. Recovery stays at or below zero ($-8\%$ and $-1\%$), confirming that the learned magnitude alone, stripped of both meaningful directions and correct category selection, cannot recover performance.
- **Random positions.** Steering at random token positions rather than SAE-gated disagreement positions collapses recovery for the smaller model ($-8\%$), while the larger model is more robust ($28\%$), indicating that position selection also contributes.

Together these confirm that both the learned category vectors and the SAE-based category and position selection are necessary, and that the coefficient MLP cannot recover performance without meaningful directions.

## 4. Related Work

**Reasoning taxonomies.** Some studies derive LLM reasoning taxonomies manually from cognitive strategies. Gandhi et al. (2025) identify four behaviors (verification, backtracking, subgoal setting, backward chaining) shared by expert humans and strong LLMs. Others derive taxonomies empirically: Marjanović et al. (2025) introduce a "thoughtology" of DeepSeek-R1, finding an optimal chain length and that excessive rumination hinders exploration. Gema et al. (2025) corroborate inverse-scaling with longer traces, with similar findings in Muennighoff et al. (2025). Sun et al. (2025) document a "ladder" of reasoning styles where even extensive fine-tuning yields diminishing returns on harder problems.

**Latent reasoning in base models.** A complementary line asks whether base models contain latent reasoning. Zhao et al. (2025) find RL post-training primarily amplifies pre-training patterns rather than teaching new skills. Wang et al. (2025) show that one carefully chosen example can markedly improve reasoning, suggesting minimal intervention can unlock latent reasoning. Minder (2024) investigates how fine-tuning surfaces pre-existing capabilities, identifying shared subspaces between base and fine-tuned models. Venhoff et al. (2025) identify interpretable activation vectors for reasoning behaviors, and Ward et al. (2025a) discover directions in base models that steer reasoning. Ward et al. (2025b) show that rank-1 LoRA adapters can recover 73–90% of reasoning performance with minimal parameter changes.

**Feature analysis.** Baek & Tegmark (2025) identify feature directions that steer different thinking modes, and Galichin et al. (2026) use SAEs to interpret reasoning features. Troitskii et al. (2025) analyze how internal features preceding "wait" tokens modulate reasoning patterns using sparse crosscoders. Bogdan et al. (2025) support sentence-level decomposition of chain-of-thought. Jia et al. (2025) integrate a learned latent action space to guide RL fine-tuning, and Zhang et al. (2025a) survey reasoning-centric LLMs.

**Inference-time steering.** Methods for steering generation without fine-tuning have gained attention. Li et al. (2025) propose Budget Guidance, modulating token probabilities to meet a target thinking budget. Fei et al. (2025) propose Nudging, framing guided decoding as inference-time alignment. Concurrently with our work, Zhang et al. (2025b) categorize reasoning into linear and non-linear behaviors, identify "cognitive heads" via linear probes, and steer these heads at test-time. Cheng et al. (2025) train lightweight probes to determine *when* to intervene and implement backtracking when deviation is detected. Both papers share our focus on identifying *when* to deploy reasoning mechanisms; our contribution differs by constructing category vectors in base models and using thinking model heuristics to probe *what* different training paradigms teach.

Our work contributes an unsupervised taxonomy of reasoning mechanisms and shows how base models can be steered along these dimensions, unifying taxonomy-driven understanding and activation-level control.

## 5. Discussion & Limitations

Our findings reveal fundamentally different training dynamics between RL and SFT-distillation. RL optimizes deployment of existing capabilities via heuristics, while SFT-distillation transfers reasoning patterns from a teacher that may be incompatible with the student's pre-existing mechanisms.

Constructive model diffing can serve as a diagnostic tool for understanding what different training paradigms teach, complementing post-hoc methods like crosscoders by directly probing whether performance differences arise from mechanism changes or heuristic learning. More broadly, this perspective may prove valuable for understanding fine-tuning beyond reasoning, including alignment, instruction-following, and domain adaptation.

Several limitations warrant future investigation. First, although we use MMLU-Pro (a multi-domain dataset) for taxonomy discovery, the resulting taxonomies are biased toward mathematical reasoning, often containing dedicated features for numerical computations. This makes evaluation on non-math benchmarks challenging; future work could develop domain-specific taxonomies or improve our discovery method to better generalize across domains.

Second, selecting appropriate steering strengths is a known challenge in interpretability. We predict per-position coefficients with a small MLP trained jointly with the category vectors on the thinking model's next-token signal. While this means the thinking model influences both the heuristic and the steering magnitude, the predicted coefficient only determines *how strongly* to steer along directions already selected by SAE classification; it does not affect *which* mechanisms fire. Nevertheless, alternative coefficient selection methods could further isolate these factors.

Finally, the lower recovery for SFT-distilled models could reflect either genuine mechanism modification or limitations in our category vector optimization. Different approaches might better capture SFT-distillation-induced changes.

## 6. Conclusion

We introduced constructive model diffing, a framework for understanding fine-tuned models by explicitly constructing the base-to-fine-tuned difference from interpretable components. For thinking models, we decompose the diff into reasoning mechanisms (category vectors) and reasoning heuristics (when to deploy each mechanism).

Our key finding is that RL-trained and SFT-distilled thinking models learn fundamentally different things. The four RL-trained models achieve roughly $76\%$ average performance recovery across GSM8K, MATH500, and a held-out Hendrycks-MATH set. The SFT-distilled models recover far less, and the mixed QwQ-32B (SFT followed by RL) is similar to the distilled models rather than to the RL-trained ones; among the distilled and mixed models recovery grows with base-model size, but even at 32B the RL-trained Open-Reasoner-Zero-32B outperforms both R1-Distill-32B and QwQ-32B. Since all model types achieve comparable benchmark performance and taxonomy quality, the gap must stem from the category vectors: they effectively induce reasoning in base models for RL settings but not, to the same degree, for SFT-based distillation. This implies that RL teaches sophisticated heuristics over pre-existing mechanisms, while SFT-based distillation additionally modifies the mechanisms themselves, with larger models able to represent more of the taught mechanisms simply because they already know more. We hope this work deepens understanding of what different training paradigms actually teach, and informs the development of more efficient methods for eliciting reasoning from base models.

## Acknowledgements

We would like to thank the ML Alignment & Theory Scholars (MATS) program for supporting this research, and in particular John Teichman and Cameron Holmes for being great research managers. We would also like to thank Chris Wendler for very helpful discussions on an early version of the work, Carolina Lucía Campi with her help on the design of the paper's main figure, and reviewers from the Mechanistic Interpretability Workshop at NeurIPS 2025 for extremely helpful feedback on early drafts of this paper.

## Author Contributions

CV did the conceptualization of the main research ideas, as well as engineering and research on the SAE taxonomy and on the many iterations of the hybrid model. CV also contributed to writing the paper. IA did engineering and research on the evaluation metrics for the SAE taxonomy, the hybrid model and significant writing for the paper. PT, AC and NN provided project advice and feedback.

## Impact Statement

This paper presents work whose goal is to advance the field of Machine Learning. There are many potential societal consequences of our work, none which we feel must be specifically highlighted here.

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

# A. SAE Training Details

Given an input vector $x \in \mathbb{R}^d$ from the residual stream and $n$ latent dimensions, a Top-K SAE learns two mappings, an encoder $f_{\text{enc}}$ and a decoder $f_{\text{dec}}$, such that:

$$z = \text{TopK}(W_{\text{enc}}(x - b_{\text{enc}})) \tag{1}$$

$$\hat{x} = W_{\text{dec}}z + b_{\text{dec}} \tag{2}$$

where $W_{\text{enc}} \in \mathbb{R}^{n \times d}$, $b_{\text{enc}} \in \mathbb{R}^n$, $W_{\text{dec}} \in \mathbb{R}^{d \times n}$, and $b_{\text{dec}} \in \mathbb{R}^d$. The training loss is then defined by the reconstruction error:

$$\mathcal{L} = \|x - \hat{x}\|_2^2 \tag{3}$$

We train our Top-K SAEs using a configuration with top-k activation sparsity where $k = 3$, meaning only the top 3 features are allowed to activate for each input. We auto-select the learning rate using the $1/\sqrt{d}$ scaling law from TinySAE (Engels, 2024): $\text{lr} = 2 \times 10^{-4}/\sqrt{n/2^{14}}$ where $n$ is the dictionary size (number of clusters), with Adam as the optimizer. Training is conducted with a batch size of 512 for a maximum of 300 epochs, implementing early stopping with a patience of 10 epochs to prevent overfitting. We apply decoder normalization after each training step, following the TinySAE implementation (Engels, 2024).

The SAEs are trained on sentence-level activations extracted from reasoning traces. We determine sentence boundaries using punctuation-based heuristics (periods, question marks, exclamation marks) and average token-level activations within each identified sentence to obtain sentence-level representations. The training data consists of 12,102 prompts from MMLU-Pro (Wang et al., 2024b), which translates into 430,122 sentences of reasoning traces from the target thinking models, where we extract activations at specific layers (6 evenly distributed layers across the model depth) and use these averaged sentence activations as inputs to the SAE training process.

The specific layers used for each model are:

- **DeepSeek-R1-Distill-Llama-8B** (32 total layers): 6, 10, 14, 18, 22, 26
- **DeepSeek-R1-Distill-Qwen-1.5B** (28 total layers): 4, 8, 12, 16, 20, 24
- **DeepSeek-R1-Distill-Qwen-14B** (48 total layers): 8, 14, 20, 26, 32, 38
- **DeepSeek-R1-Distill-Qwen-32B** (64 total layers): 9, 18, 27, 36, 45, 54
- **QwQ-32B** (64 total layers): 9, 18, 27, 36, 45, 54
- **Open-Reasoner-Zero-0.5B** (24 total layers): 5, 8, 11, 14, 17, 20
- **Open-Reasoner-Zero-1.5B** (28 total layers): 4, 8, 12, 16, 20, 24
- **Open-Reasoner-Zero-7B** (28 total layers): 4, 8, 12, 16, 20, 24
- **Open-Reasoner-Zero-32B** (64 total layers): 9, 18, 27, 36, 45, 54

## A.1. Feature Stability Across Random Seeds

*Table 3.* **SAE feature stability across random seeds** for the ORZ-7B SAE (layer 16, dictionary size 15). Features are matched one-to-one via the Hungarian algorithm on cosine similarity; a pair counts as matched when cosine $> 0.7$ (Paulo & Belrose, 2026).

| Seed pair | Matched ($> 0.7$) | Mean cosine |
|---|---|---|
| seed1 vs seed2 | 80.0% | 0.850 |
| seed1 vs seed3 | 93.3% | 0.834 |
| seed1 vs seed4 | 93.3% | 0.894 |
| seed1 vs seed5 | 93.3% | 0.837 |
| seed2 vs seed3 | 93.3% | 0.895 |
| seed2 vs seed4 | 86.7% | 0.864 |
| seed2 vs seed5 | 93.3% | 0.892 |
| seed3 vs seed4 | 100.0% | 0.909 |
| seed3 vs seed5 | 100.0% | 0.972 |
| seed4 vs seed5 | 100.0% | 0.911 |
| **Overall** | **93.3%** | **0.886** |

To assess whether our discovered features reflect stable structure rather than seed-induced artifacts, we replicate the protocol of Paulo & Belrose (2026). We retrain an ORZ-7B SAE (layer 16, dictionary size 15) five times with seeds $\{1, 2, 3, 4, 5\}$.

The runs exhibit genuine variety in convergence dynamics, with early stopping occurring between epochs 17 and 35 and final losses in the range 0.693–0.695, confirming the seeds produce meaningfully different training trajectories. For each pair of runs, we match the 15 features one-to-one using the Hungarian algorithm on cosine similarity and, following Paulo & Belrose (2026), label a pair as "matched" when its cosine similarity exceeds 0.7. As shown in Table 3, we obtain an average matching rate of 93.3% and a mean pairwise cosine of 0.886 across the ten comparisons, substantially above the $\sim 30\%$ reported for large SAEs and consistent with the finding that smaller models and dictionaries share more features.

## B. Details of Taxonomy Evaluation

### B.1. Cluster Title & Description Generation

We use OpenAI's o4-mini model to generate the cluster title and description.

Concretely, we prompt this model to carefully look at the examples and identify the shared reasoning strategy or cognitive mechanism, common linguistic patterns or structures, specific phrases or words common to the category, and the functional role within the overall reasoning process. The model then produces a concise title naming the specific reasoning function and a detailed description that explains what the function does, what is included, and what is excluded from this category. This prompt is shown below:

```
1  Analyze the following [N] sentences from an LLM reasoning trace. These sentences are grouped into a cluster
       based on their similar role or function in the reasoning process.
2
3  Your task is to identify the precise cognitive function these sentences serve in the reasoning process. Consider
       the reasoning strategy or cognitive operation being performed.
4
5  Sentences:
6  '''
7  [LIST OF EXAMPLE SENTENCES]
8  '''
9
10 Look for:
11 - Shared reasoning strategies or cognitive mechanisms
12 - Common linguistic patterns or structures
13 - Functional role within the overall reasoning process
14
15 [OPTIONAL: CATEGORY EXAMPLES SECTION WITH 5 EXAMPLE CATEGORIES]
16
17 Your response should be in this exact format:
18 Title: [crisp, single-concept title without slashes, parentheses, or compound phrases]
19 Description: [3-4 sentences explaining (1) the specific reasoning process this cluster represents, (2) what is
       INCLUDED in this category, (3) what is NOT INCLUDED in this category]
20
21 Guidelines for titles:
22 - Use simple, clear nouns or verb phrases
23 - Avoid slashes (/) and parentheses ()
24 - Capture one core reasoning concept
25
26 Guidelines for descriptions:
27 - Focus on the specific cognitive or reasoning function
28 - Avoid abstracting too much from the specific examples
29 - Mention specific phrases or words that are common in the examples
30 - Be precise enough that someone could reliably identify new examples of this reasoning function.
31
32 In summary, the description should be as sharp and specific as possible, and the title should be as simple and
       abstract as possible.
```

*Prompt 1.* Prompt used for generating cluster descriptions and titles

### B.2. Consistency (F1 Score)

To evaluate how well our categories can reliably classify individual sentences, we implement a binary classification task. For each category, we sample example sentences from within the category (positive examples) and from outside the category (negative examples). An LLM-based autograder (OpenAI's GPT-4.1-mini) receives the category title and description along with these examples and must classify each as either belonging to the category or not. We calculate precision, recall, and F1 scores for each category, then take the average F1 score across all categories as our overall consistency score. The complete prompt is shown below:

```
1  # Task: Binary Classification of Reasoning Sentences by Function
2
```

```
 3 You are an expert at analyzing the *function* of sentences within a longer chain of reasoning. Your task is to
       determine if each sentence below performs the specific cognitive or procedural role described.
 4
 5 **Core Principle:** Do not focus on the surface-level topic of the sentence. Instead, abstract away from the
       specific content and ask: "What *job* is this sentence doing in the reasoning trace?"
 6
 7 ## Category Description:
 8 Title: [TITLE]
 9 Description: [DESCRIPTION]
10
11 ## Sentences to Classify:
12 [FORMATTED SENTENCES]
13
14 ## Instructions:
15 1. For each sentence, identify its functional role in a potential reasoning process.
16 2. Compare this role to the category description provided.
17 3. If the sentence's function matches the description, assign "Yes". Importantly, a sentence might not match a
       description word-for-word, but it might serve the same underlying purpose.
18 4. If the sentence's function does not align with the category, assign it "No".
19 5. Respond with "Yes" or "No" for each sentence.
20
21 ## Response Format:
22 Your response must follow this exact JSON format:
23 ```json
24 {
25   "classifications": [
26     {
27       "sentence_id": <sentence idx>,
28       "belongs_to_category": "Yes" or "No",
29       "explanation": "Brief explanation of your reasoning"
30     }
31   ]
32 }
33 ```
34
35 Only include the JSON object in your response, with no additional text before or after.
```

*Prompt 2.* Prompt used for the F1 score (accuracy) autograder

### B.3. Completeness (Confidence Score)

We evaluate how well individual sentences fit their assigned categories by having an LLM (GPT-4.1-mini) rate the quality of each assignment on a scale from 0-10. This measures the confidence in our category assignments and serves as our completeness metric. The scores are afterwards normalized to a 0-1 scale for compatibility with the final score calculation. The complete prompt is shown below:

```
 1 You are an expert at analyzing how well individual sentences match their assigned reasoning function categories.
       Your task is to evaluate how well a given sentence exemplifies the specific cognitive or procedural role
       described in its assigned category.
 2
 3 # Sentence to Evaluate:
 4 [SENTENCE]
 5
 6 # Assigned Category:
 7 Title: [TITLE]
 8 Description: [DESCRIPTION]
 9
10 # Instructions:
11 1. Carefully analyze the sentence's content and the functional role it might display in a reasoning process.
12 2. Compare this content and role to the category description provided.
13 3. Consider how well the sentence matches the category description.
14 4. Provide a brief explanation of your reasoning.
15 5. Rate the fit on a scale from 0-10, where:
16    - 0 = Very poor fit, sentence does not match the category at all
17    - 10 = Perfect fit, sentence matches exactly the category description
18
19 # Response Format:
20 Your response must follow this exact JSON format. The explanation must be a single-line string with no newlines:
21 ```json
22 {
23   "explanation": "Brief explanation of how well the sentence matches the category and your reasoning for the
       score",
24   "completeness_score": <integer from 0-10>
25 }
26 ```
27
```

```
28 Only include the JSON object in your response, with no additional text before or after.
```

*Prompt 3.* Prompt used for the completeness autograder

## B.4. Independence (Semantic Orthogonality)

To ensure that our taxonomy categories represent functionally distinct reasoning mechanisms, we evaluate the semantic similarity between all pairs of categories using an LLM-based approach. For each pair of categories in a cluster, an LLM (GPT-4.1-mini) evaluates how similar they are in terms of their underlying cognitive or functional purpose on a scale from 0-10, where 0 means completely different reasoning functions and 10 means essentially the same function. We then calculate the semantic orthogonality score as the fraction of category pairs that have an orthogonality score above a threshold $(0.5)$, where orthogonality is defined as $1 - $ similarity, indicating functional independence between categories. The complete prompt is shown below:

```
 1 # Task: Semantic Similarity Evaluation
 2
 3 You are an expert at analyzing the semantic similarity between different reasoning functions. Your task is to
      evaluate how similar two categories of reasoning sentences are in terms of their underlying cognitive or
      functional purpose.
 4
 5 ## Category 1:
 6 Title: [TITLE1]
 7 Description: [DESCRIPTION1]
 8
 9 ## Category 2:
10 Title: [TITLE2]
11 Description: [DESCRIPTION2]
12
13 ## Instructions:
14 Rate the semantic similarity between these two categories on a scale from 0 to 10, where:
15 - 0 = Completely different reasoning functions
16 - 5 = Somewhat related but distinct functions
17 - 10 = Essentially the same reasoning function, just described differently
18
19 Consider:
20 1. The underlying cognitive process or reasoning operation
21 2. The functional role within a reasoning trace
22 3. Whether sentences from one category could reasonably belong to the other
23
24 Focus on functional similarity rather than surface-level word overlap.
25
26 ## Response Format:
27 Your response must follow this exact JSON format:
28 ```json
29 {
30   "explanation": "Brief explanation of your reasoning for this score",
31   "similarity_score": <integer from 0-10>
32 }
33 ```
34
35 Only include the JSON object in your response, with no additional text before or after.
```

*Prompt 4.* Prompt used for the semantic orthogonality evaluation

## B.5. Decoder Weight Vector Orthogonality

The independence of our taxonomy can also be measured by the orthogonality between the decoder latents (centroids) in our Sparse Autoencoder (SAE). For a set of decoder weight vectors $\{w_1, w_2, ..., w_n\}$ where $n$ is the number of categories, we calculate:

$$\text{Similarity}_{i,j} = \frac{w_i \cdot w_j}{\|w_i\| \cdot \|w_j\|} \tag{4}$$

This produces a cosine similarity matrix where values close to $0$ indicate nearly orthogonal (independent) features. We then compute:

- The average absolute cosine similarity between all pairs of latents
- The maximum absolute cosine similarity between any pair of latents

Lower values for both metrics indicate better independence between our taxonomy categories. This orthogonality analysis ensures that our categories represent distinct reasoning mechanisms rather than variations of the same underlying process.

However, in practice, we found that these cosine similarity values were consistently very high (near 1.0) across different SAE configurations, providing limited discriminative power for comparing different taxonomies. This led us to adopt the semantic orthogonality metric instead, which better captures functional distinctness between reasoning categories.

### B.6. Choice of LLM Models for Taxonomy Evaluation

We employ different LLMs for different evaluation tasks based on their computational requirements and criticality to downstream performance. Category title and description generation uses OpenAI's o4-mini, a more sophisticated reasoning model, because these titles fundamentally determine the semantic boundaries of each category and directly impact all subsequent evaluation metrics. In contrast, consistency, completeness, and independence evaluations use GPT-4.1-mini, a capable but more cost-effective model, as these tasks involve more straightforward classification and rating given well-defined categories. This design choice is further motivated by our evaluation scale: we generate 5 repetitions of category titles for each configuration across our extensive grid search, making the computational cost of using premium models for all evaluation steps prohibitive while maintaining evaluation quality where it matters most.

### B.7. Scoring Normalization

For our grid search visualization and comparison across different configurations, we normalize each metric to a 0-1 scale using min-max normalization within each model. This normalization is performed across all layer and cluster size combinations for a given model, ensuring that the final normalized score reflects relative performance within each model's configuration space. The normalization formula is:

$$\text{Normalized Score} = \frac{\text{Raw Score} - \text{Min Score}}{\text{Max Score} - \text{Min Score}} \tag{5}$$

where Min Score and Max Score are computed across all layer/cluster combinations for a single model.

## C. Human Validation of LLM Judges

We conducted a human evaluation of every LLM judge used in the paper (400 annotations in total).

**Benchmark scoring judge.** The LLM judge that produces all hybrid-evaluation accuracies in Table 1 achieves Cohen's $\kappa = 0.880$ against human grading, validating our central empirical results.

**Taxonomy grading judges.** For the consistency, completeness, and independence judges (Appendices B.2 to B.4), human–judge agreement falls in the $\kappa/r = 0.45$–$0.52$ range overall, with disagreement concentrated in genuinely ambiguous categories; on well-defined categories agreement rises to $\kappa = 0.800$ and $r = 0.918$.

**Are the category labels meaningful?** To test the labels directly, a human annotator wrote a title and description for each of the 15 ORZ-7B categories from the raw SAE cluster exemplars, without seeing the LLM-generated labels. Comparing human and LLM labels yields $53\%$ (8/15) exact matches, $40\%$ (6/15) partial matches, and $7\%$ (1/15) genuine disagreement. The partial matches and the single disagreement are informative: the human tended to describe the content domain (e.g. "Calculate physical properties"), whereas the LLM captured the higher-level cognitive mechanism (e.g. "Applying Known Formulas"). The lone disagreement follows the same pattern ("Logical & Combinatorial Statements" for the human vs. "Stating Intermediate Results" for the LLM), where the LLM label better reflects the reasoning function relevant to our pipeline. Overall, 14/15 categories show full or partial agreement with the human annotator, which we take as strong validation of category quality.

## D. Qualitative Example of Hybrid Steering

Figure 5 shows a representative example of the hybrid model solving a MATH500 problem, with each steered token position colored by the SAE reasoning category that triggered steering. Notably, the steering induces metacognitive behaviour: after an initial numerical estimate the hybrid model verifies its own result, judges it to be an underestimate, and then explores concrete candidate values to recover the true maximum, whereas the unsteered base model commits to an incorrect answer.

**Hybrid Model in Action: ORZ-32B**
Qwen2.5-32B base · Open-Reasoner-Zero-32B thinking vectors · MATH500

Let a, b, c be real numbers such that |ax^2 + bx + c| ≤ 1for all 0 ≤ x ≤ 1. Find the largest possible value of |a| + |b| + |c|.

| | | |
|---|---|---|
| **Correct answer: 17** | **Base model: incorrect** | **Hybrid model: correct** |

*Hybrid model response (steered token positions highlighted by reasoning category):*

Let's start by understanding the given condition: |ax^2 + bx + c| ≤ 1 for all 0 ≤ x ≤ 1 . This means that the quadratic function f(x) = ax^2 + bx + c must lie between -1 and 1 for all x in the interval [0, 1].

To find the largest possible value of |a| + |b| + |c| , we need to consider the values of f(x) at the endpoints and the midpoint of the interval [0, 1]. Specifically, we need to evaluate f(0) , f(1) , and f((1)/(2)) .

Let's break down the problem step-by-step.

Step 1: Evaluate f(0) , f(1) , and f((1)/(2))
We have:
f(0) = c
f(1) = a + b + c
f((1)/(2)) = (a)/(4) + (b)/(2) + c

Since |f(x)| ≤ 1 for all 0 ≤ x ≤ 1 , we have:
|c| ≤ 1
|a + b + c| ≤ 1
| (a)/(4) + (b)/(2) + c | ≤ 1

Step 2: Express a , b , and c in terms of the constraints
We need to maximize |a| + |b| + |c| . To do this, we can consider the extreme values of a , b , and c that satisfy the given constraints.

Let's break down the constraints further:
1. |c| ≤ 1
2. |a + b + c| ≤ 1
3. | (a)/(4) + (b)/(2) + c | ≤ 1

We can use these constraints to find the maximum possible value of |a| + |b| + |c| .

Step 3: Solve the system of inequalities
To find the maximum value, we can use the method of Lagrange multipliers or simply analyze the constraints. Let's break it down into manageable parts and use Python to find the solution.

Let's write the code to find the maximum value of |a| + |b| + |c| under the given constraints.
[runs code]
[tool output]
The optimization result indicates that the maximum value of |a| + |b| + |c| is 1. However, this seems to be an underestimation. Let's manually verify the constraints and see if we can find a better solution.

Step 4: Manual Verification
Let's consider the constraints again:
1. |c| ≤ 1
2. |a + b + c| ≤ 1
3. | (a)/(4) + (b)/(2) + c | ≤ 1

We need to find values of a , b , and c that maximize |a| + |b| + |c| . Let's try some specific values that satisfy the constraints.

For example, let's consider a = 8 , b = -8 , and c = 1 :
- |c| = 1 ≤ 1
- |a + b + c| = |8 - 8 + 1| = 1 ≤ 1
- | (a)/(4) + (b)/(2) + c | = | (8)/(4) + (-8)/(2) + 1 | = |2 - 4 + 1| = 1 ≤ 1

These values satisfy all the constraints, and |a| + |b| + |c| = 8 + 8 + 1 = 17 .

Let's check if there are any other values that might give a higher sum. We can try different combinations, but it seems that the values a = 8 , b = -8 , and c = 1 give us the maximum sum.

Therefore, the largest possible value of |a| + |b| + |c| is 17.

Reasoning categories (steered token positions only)

| | | |
|---|---|---|
| ■ Performing Calculations | ■ Problem Decomposition | ■ Generating Candidate Answers |
| ■ Deducing Quantitative Relationships | ■ Structuring Problem-Solving Steps | ■ Numeric Calculation Steps |

*Figure 5.* **Hybrid model in action.** Example of a hybrid model (Qwen2.5-32B as base model with category vectors trained on Open-Reasoner-Zero-32B) solving a MATH500 problem. Each token in the model's response is highlighted according to the SAE category that triggered steering at that position. The model is steered at ~12% of token positions, spanning six distinct reasoning categories (Performing Calculations and Problem Decomposition most prominent). After an initial estimate it self-verifies ("this seems to be an underestimation. Let's manually verify . . . and see if we can find a better solution"), then searches over candidate values and arrives at the correct answer 17, while the unsteered base model fails.

# E. Details of Hybrid Model Evaluation

## E.1. Training Data

We construct a diverse training mixture of 9,794 questions (with a held-out validation set of 995 questions) drawn from four sources: MATH (Hendrycks et al., 2021) (5,000 train / 500 val), Natural Reasoning (Yuan et al., 2025) (3,500 / 350), SciBench (Wang et al., 2024a) (622 / 70), and TheoremQA (Chen et al., 2023) (672 / 75). The MATH questions used for training are drawn from the Hendrycks-MATH training split and are strictly disjoint from the MATH500 benchmark: none of the 500 MATH500 problems appear in either the training or validation portion of our mix, so MATH500 remains a genuine out-of-sample evaluation. The mix is designed to ensure that every SAE-discovered reasoning category has sufficient training signal: mathematical datasets provide training data for arithmetic, algebraic, and formula-related categories, while Natural Reasoning and TheoremQA cover broader reasoning patterns such as task formulation, definitions, and inference steps. For each question, we generate a thinking-model rollout using the appropriate thinking model served via vLLM (Kwon et al., 2023) (temperature 0.6, top-$p$ 0.95, max 2,048 tokens), producing reasoning traces that the category vectors are trained to reproduce.

**Evaluation data.** We evaluate the hybrid models on three benchmarks: GSM8K (Cobbe et al., 2021), MATH500 (Hendrycks et al., 2021), and an additional Hendrycks-MATH holdout of 1,000 questions. The Hendrycks-MATH holdout is sampled from the Hendrycks-MATH pool and is constructed to be disjoint from both the category-vector training mix and MATH500 (zero shared problems), providing a second, larger in-distribution mathematical benchmark that stress-tests generalization beyond the 500-question MATH500 set.

## E.2. Category Vector Training

**Disagreement collection.** Given a (base, thinking) model pair, we identify training positions via teacher-forced disagreement: the base model processes the thinking model's rollout, and at each token position we check whether the base model's greedy prediction matches the ground-truth next token from the thinking rollout. Positions where the base model disagrees are candidate steering targets. Each disagreement position is labelled with a reasoning category by running the thinking model's hidden state at the SAE layer through the SAE encoder and taking the $\arg\max$ over latent activations. To keep training balanced, we cap positions at 64 per (example, category) pair.

**Optimization.** We jointly optimize two components: a matrix of per-category vectors $V \in \mathbb{R}^{K \times d}$ (where $K$ is the number of SAE categories and $d$ is the model's hidden dimension) and a small MLP that predicts a non-negative steering coefficient $\alpha$ for each position. The MLP consists of a shared trunk (`Linear`$(d, 512) \to$ `GELU`) followed by $K$ per-category heads (`Linear`$(512, 1) \to$ `Softplus`), so that the applied shift at position $p$ with category $c$ is $\alpha(h_p, c) \cdot V_c$, where $h_p$ is the base model's residual-stream activation at the steering layer.

The training loss is cross-entropy on the ground-truth next token, computed only at disagreement positions under the steered base model. To prevent categories with more positions from dominating the gradient, we compute the per-category mean loss and then average across categories present in each minibatch. We use AdamW with separate learning rates for the category vectors ($10^{-2}$) and MLP parameters ($10^{-3}$), weight decay 0.01, and gradient clipping at norm 1.0 for the MLP.

**Validation and early stopping.** We evaluate after each epoch on three held-out sets: (1) the 995-question validation split of the training mix, (2) 500 MATH500 questions (Hendrycks et al., 2021), and (3) 500 GSM8K questions (Cobbe et al., 2021), the latter two serving as true out-of-sample benchmarks not seen during training. The primary selection metric is per-sample cross-entropy on the validation split. Each category vector is saved at its individually best epoch, and training terminates when no category has improved for 5 consecutive epochs. We train for up to 10 epochs with a batch size of 4.

**Best-of-three selection via the holdout mix.** Category-vector training has non-trivial run-to-run variance, so for every model pair we train three independent sets of category vectors (and their coefficient MLPs) with different random seeds. To choose among them without touching the evaluation benchmarks, we build a hybrid model from each of the three sets and measure its gap recovered on the "holdout mix": a gold-answer subset (788 questions) of the training-mix validation split that is never used for gradient updates. Concretely, for each set we run the full hybrid construction on the holdout mix and compute the gap recovered. We select the vector set with the highest holdout-mix gap recovered and use only that set for all downstream hybrid evaluations reported in this paper (GSM8K, MATH500, and the Hendrycks-MATH

holdout). Because selection uses the training-mix validation split rather than any evaluation benchmark, the reported GSM8K/MATH500/Hendrycks-MATH numbers remain out-of-sample.

### E.3. Prompt Templates

At both training and evaluation time, base models receive a simple question–answer completion prompt with no chat template applied:

```
Answer the following question:
Q: [question]
A:
```

We deliberately use this minimal question–answer prompt so that any reasoning behaviour exhibited by the base model is induced by the SAE-selected category steering vectors, not seeded by the prompt.

For thinking models, prompting follows each model family's official inference guidelines. Since all of our benchmarks are mathematical, every thinking model additionally receives the DeepSeek-R1/QwQ step-by-step directive ("Please reason step by step, and put your final answer within \boxed{}.") appended after the question. Open-Reasoner-Zero (ORZ) models use the Table 5 template from Hu et al. (2025), which prepends an answer-format instruction and, on these math benchmarks, also appends the step-by-step directive:

```
You must put your answer inside <answer> </answer> tags,
i.e., <answer> answer here </answer>. And your final
answer will be extracted automatically by the \boxed{} tag.
[question]

Please reason step by step, and put your final answer
within \boxed{}.
```

This is wrapped in the model's native chat template, which appends the <think> opening tag. For DeepSeek-R1-Distill and QwQ models, we append only the recommended reasoning directive after the question (Guo et al., 2025; Qwen Team, 2024):

```
[question]

Please reason step by step, and put your final answer
within \boxed{}.
```

These are processed via each model's native chat template. Both base and hybrid models use identical prompting at evaluation time, ensuring that performance differences are not attributable to prompt formatting.

## F. Category Vector Usage Statistics

Table 4 lists the three most frequently activated reasoning categories per pair, while Table 5 reports how the hybrid model applies category vectors: the fraction of steered tokens (mean $\pm$ std across problems) and the MLP-predicted coefficient $\alpha$ (mean $\pm$ std over all steered positions).

*Table 4.* **Top-3 activated reasoning mechanisms per model pair.** For each pair we list the three SAE categories most frequently triggered during steering, with their share (%) of total steered positions.

| Base Model | Thinking Model | Bench. | Top-3 Activated Mechanisms (% of steered positions) |
|---|---|---|---|
| *SFT-Distilled Thinking Models* | | | |
| Qwen2.5-Math-1.5B | R1-Distill-1.5B | GSM8K | Numeric Computation (22), Metacognitive Self-Prompts (20), Identifying Additional Factors (14) |
| | | MATH500 | Metacognitive Self-Prompts (23), Numeric Computation (18), Identifying Additional Factors (16) |
| Llama-3.1-8B | R1-Distill-8B | GSM8K | Restating the Question (28), Arithmetic Computation (22), Calculation step (12) |
| | | MATH500 | Restating the Question (23), Calculation step (19), Arithmetic Computation (18) |
| Qwen2.5-14B | R1-Distill-14B | GSM8K | Problem Restatement (65), Metacognitive Markers (35), Conditional Reasoning ($<$1) |
| | | MATH500 | Problem Restatement (82), Metacognitive Markers (17), Numeric Computation ($<$1) |
| Qwen2.5-32B | R1-Distill-32B | GSM8K | Problem Framing (39), Evaluating Expressions (24), Stating Given Problem Data (9) |
| | | MATH500 | Evaluating Expressions (41), Problem Framing (22), Computational step initiation (15) |
| *RL-Trained Thinking Models* | | | |
| Qwen2.5-0.5B | ORZ-0.5B | GSM8K | Algebraic Manipulation Steps (32), Answer Declarations (32), Arithmetic Computation (7) |
| | | MATH500 | Algebraic Manipulation Steps (36), Answer Declarations (28), Symbolic Derivation (16) |
| Qwen2.5-1.5B | ORZ-1.5B | GSM8K | Answer Declaration (18), Functional Explanation (17), Step-by-Step Planning (16) |
| | | MATH500 | Substitution and Computation (19), Functional Explanation (17), Step-by-Step Planning (17) |
| Qwen2.5-7B | ORZ-7B | GSM8K | Performing Calculation Steps (29), Listing Given Data (23), Stepwise Analysis (17) |
| | | MATH500 | Stepwise Analysis (29), Performing Calculation Steps (25), Calculation Steps (15) |
| Qwen2.5-32B | ORZ-32B | GSM8K | Numeric Calculation Steps (21), Performing Calculations (20), Problem Decomposition (14) |
| | | MATH500 | Performing Calculations (37), Numeric Calculation Steps (12), Case-by-Case Evaluation (9) |
| *Mixed (SFT + RL)* | | | |
| Qwen2.5-32B | QwQ-32B | GSM8K | Arithmetic Computation Steps (29), Speculating answer choices (17), Meta-Cognitive Check-ins (11) |
| | | MATH500 | Arithmetic Computation Steps (27), Meta-Cognitive Check-ins (18), Recall of Standard Equations (16) |

*Table 5.* **Category vector usage statistics.** Fraction of generated tokens receiving a steering intervention and the MLP-predicted coefficient $\alpha$, reported as mean $\pm$ std across problems.

| | | GSM8K | | MATH500 | | Hendrycks-MATH | |
|---|---|---|---|---|---|---|---|
| **Base Model** | **Thinking Model** | **Steered (%)** | **MLP $\alpha$** | **Steered (%)** | **MLP $\alpha$** | **Steered (%)** | **MLP $\alpha$** |
| *SFT-Distilled Thinking Models* | | | | | | | |
| Qwen2.5-Math-1.5B | R1-Distill-1.5B | $15.8 \pm 8.6$ | $3.55 \pm 4.03$ | $12.0 \pm 5.5$ | $3.96 \pm 4.35$ | $12.1 \pm 6.0$ | $3.99 \pm 4.41$ |
| Llama-3.1-8B | R1-Distill-8B | $17.6 \pm 24.1$ | $0.25 \pm 0.16$ | $12.5 \pm 15.9$ | $0.26 \pm 0.16$ | $13.4 \pm 17.7$ | $0.26 \pm 0.16$ |
| Qwen2.5-14B | R1-Distill-14B | $15.0 \pm 10.8$ | $2.25 \pm 1.76$ | $12.9 \pm 8.2$ | $2.23 \pm 1.71$ | $13.5 \pm 9.4$ | $2.29 \pm 1.74$ |
| Qwen2.5-32B | R1-Distill-32B | $23.3 \pm 16.5$ | $3.37 \pm 4.77$ | $23.4 \pm 22.0$ | $5.02 \pm 7.57$ | $23.7 \pm 22.7$ | $5.06 \pm 7.64$ |
| *RL-Trained Thinking Models* | | | | | | | |
| Qwen2.5-0.5B | ORZ-0.5B | $7.1 \pm 8.7$ | $0.81 \pm 0.68$ | $5.9 \pm 9.1$ | $0.74 \pm 0.73$ | $5.3 \pm 7.7$ | $0.74 \pm 0.73$ |
| Qwen2.5-1.5B | ORZ-1.5B | $8.8 \pm 4.7$ | $1.41 \pm 0.93$ | $4.9 \pm 3.5$ | $1.19 \pm 0.96$ | $5.0 \pm 4.0$ | $1.21 \pm 0.98$ |
| Qwen2.5-7B | ORZ-7B | $6.9 \pm 2.5$ | $1.77 \pm 1.45$ | $5.9 \pm 2.4$ | $1.55 \pm 1.48$ | $6.0 \pm 2.6$ | $1.59 \pm 1.50$ |
| Qwen2.5-32B | ORZ-32B | $11.5 \pm 3.3$ | $11.28 \pm 8.47$ | $12.2 \pm 5.5$ | $9.95 \pm 7.34$ | $12.2 \pm 5.6$ | $9.92 \pm 7.28$ |
| *Mixed (SFT + RL)* | | | | | | | |
| Qwen2.5-32B | QwQ-32B | $28.6 \pm 24.6$ | $5.09 \pm 4.92$ | $22.7 \pm 20.5$ | $6.71 \pm 6.12$ | $22.0 \pm 18.6$ | $6.66 \pm 6.06$ |

# G. Sparse Autoencoder Features

In this section, we provide detailed tables showing the complete reasoning taxonomies for each of the nine model pairs evaluated in our experiments. For each model we list all category titles, the full description of what each feature captures, and a representative top-activating example sentence drawn from the reasoning traces.

## G.1. Open-Reasoner-Zero-0.5B (Layer 8, Dict Size 10)

*Table 6.* Reasoning taxonomy for Open-Reasoner-Zero-0.5B (Layer 8, Dict Size 10). Categories are identified by our SAE; examples are top-activating sentences.

| Category | Description | Top-Activating Example |
|---|---|---|
| Algebraic Manipulation Steps | This category covers the sentences that explicitly perform or introduce a symbolic or numerical operation on an equation or expression – substituting values, simplifying fractions, rearranging terms, solving for a variable, summing or factoring, and rounding intermediate results. These moves are signaled by phrases like "Substituting....," "Simplifying....," "Therefore, we have....," "Rearranging to solve for....," or "Plugging in these values...." They serve to carry out the concrete computational transformations in a step-by-step derivation. This category does not include higher-level planning, assumption-stating, recall of definitions, or expressions of uncertainty. | "Therefore, we can simplify the expression:" |
| Arithmetic Computation | This cluster consists of sentences that carry out explicit numeric operations – unit conversions, multiplications, divisions, additions, or subtractions – to produce intermediate quantitative values. Included are steps like "632.8 nm $= 6.328 \times 10^{-7}$ m," "44 hours + 4 hours + 3 hours $= 51$ hours," and "80000 ft$^3$ $\times$ 0.0283168 m$^3$/ft$^3$ $= 22441.36$ m$^3$." Not included are sentences that propose hypotheses, express uncertainty, outline future reasoning steps, or recall definitions; instead they focus solely on the mechanical arithmetic needed to advance the calculation. | "The wind tunnel operates at a pressure of 20 atmospheres, which is equivalent to $20 \times 101325$ Pa $= 2026500$ Pa." |

*Continued on next page*

*Continued from previous page*

| Category | Description | Top-Activating Example |
|---|---|---|
| **Symbol and Term Definitions** | These sentences establish the precise meaning of symbols, variables, predicates, or specialized terms before they are used in the argument. They include statements like "$Z$ is the atomic number," "$P(x)$ denotes …," "a metric space $Y$ is bounded if …," or simple value bindings such as "The tip is 20% of the lunch cost." They function to ground the ensuing reasoning by spelling out each symbol's referent or each term's formal definition. This category does not cover steps of deduction, calculation procedures, speculative hypotheses, uncertainty acknowledgments, or future-planning statements. | "**Carbon (C)**: Carbon is a non-metal." |
| **Legal Findings and Conclusions** | This cluster comprises sentences where the reasoner states the outcome of applying legal rules to facts – either factual findings, procedural determinations, or normative judgments ("the court found…," "the judge would conclude…," "the best defense is…," "the court should dismiss…"). These lines articulate definitive legal decisions or predicted rulings rather than listing evidence, proposing next steps, or expressing uncertainty. They are distinguished by verbs of judgment (find, conclude, determine, instruct, dismiss, grant, deny) and present the resolved legal stance. Sentences that merely describe raw evidence, raise questions, outline plans, or express tentativeness are not included. | "The buyer sued the seller for breach of warranty, and the seller has asked an attorney to advise whether the parties' contract requires arbitration of the buyer's claim." |
| **Answer Declarations** | These sentences serve to present the final outcome of a reasoning step by explicitly stating the computed value, selected option, diagnosis, or solution. They typically begin with markers like "<answer>," "The answer is," "Therefore, the answer is," or are formatted as a boxed result. This category includes only statements that deliver a conclusion without further justification, intermediate calculation, or uncertainty. It does not include planning steps, hypothesis generation, assumption statements, definitions, or expressions of uncertainty. | "<answer> Therefore, the most prevalent mental disorder in the world is $\boxed{schizophrenia}$." |
| **Symbolic Derivation** | This category captures the steps where the reasoner writes down or transforms mathematical expressions and equations as part of a calculation. Included are instances of recalling standard formulas, applying algebraic identities, isolating variables, performing substitutions, and computing intermediate numeric or symbolic results (e.g., "use the quadratic formula…," "P(X=k)=…," "$(9+d)^2$=(9+2d)·29"). Not included are any explanations of why a particular formula is chosen, meta-level planning, uncertainty acknowledgments, or high-level conceptual reasoning – these sentences are strictly the raw symbolic or algebraic manipulations. | $f_b = \left\lvert \frac{c}{\lambda} - \frac{c}{V} \right\rvert$ |
| **Stating Known Equations** | This cluster consists of sentences that retrieve or assert established mathematical or physics relationships without applying them yet. Each sentence introduces a formula or definitional link – for example, "The angular frequency $\omega$ is given by $\omega = 2\pi f$," or "The mass flow rate $\dot{m}$ is the product of volume flow rate Q and density $\rho$." Included are canonical statements signaled by phrases like "is given by," "is equal to," "is related to," or "is the product of." Not included are tentative plans, uncertainty remarks, assumption declarations, or speculative hypotheses. | "The average heat flux $Q_{\text{avg}}$ is the heat transfer rate per unit area divided by the area of the plate:" |

| Category | Description | Top-Activating Example |
|---|---|---|
| **Action Step Specification** | These sentences explicitly lay out the next operation or subtask in the solution, using imperatives or numbered/ordered cues to sequence the reasoning. They typically begin with verbs like "Determine," "Calculate," "Assess," or markers such as "Next," "Step 1:," "Step 2:," to announce what to do without executing it. They serve to structure the argument into discrete, actionable steps toward the answer. This category does not include statements of definitions, assumptions, uncertainty acknowledgments, or explanatory hypotheses. | **"Determine the vacation liability:"** |
| **Background Knowledge Statements** | These sentences supply general, domain-specific facts, definitions, or properties needed to ground the chain of thought. They include declarative statements that define concepts ("The goal of psychotherapy is…," "Supply Chain Integration means…"), describe system components ("They are located in the retina…"), or outline characteristic roles and functions in various fields. They do not plan future steps, express uncertainty, propose hypotheses, or perform calculations, but instead provide the conceptual groundwork on which subsequent reasoning builds. | "This diversity can lead to a more complex and diverse community, which can have significant impacts on the ecosystem's resilience and stability." |
| **Articulating Subtasks** | These sentences lay out the immediate conceptual or procedural prerequisites needed to tackle the problem, typically beginning with "To … we need to …." and specifying which concept to understand, factor to consider, or data to gather next. They function as planning or decomposition steps, breaking the overall task into clear subgoals without yet performing calculations, retrieving definitions, or testing hypotheses. They do not include detailed execution, factual recall, uncertainty acknowledgments, or speculative explanations. | "To solve the problem of determining the relationship between a motor and a generator, we need to understand the basic components and their functions." |

## G.2. Open-Reasoner-Zero-1.5B (Layer 4, Dict Size 10)

*Table 7.* Reasoning taxonomy for Open-Reasoner-Zero-1.5B (Layer 4, Dict Size 10). Categories are identified by our SAE; examples are top-activating sentences.

| Category | Description | Top-Activating Example |
|---|---|---|
| **Step-by-Step Planning** | These sentences introduce and outline the procedural plan for solving a problem by specifying that a sequence of actions will be taken. They commonly begin with infinitive or imperative phrases – "To determine … we need to follow these steps," "To find … start by," "Here's how … works" – and serve as signposts for the forthcoming multi-step solution. Included are any statements that announce but do not yet perform calculations, that lay out the structure of reasoning. Not included are sentences that actually carry out computations, recall definitions, state assumptions, propose hypotheses, or express uncertainty. | "To determine the most likely decision of the prosecutor, we need to analyze the situation step by step:" |

*Continued from previous page*

| Category | Description | Top-Activating Example |
|---|---|---|
| **Substitution and Computation** | This cluster comprises the procedural execution steps where the reasoner directly applies formulas by inserting given numbers or expressions, simplifies algebraic terms, and carries out basic arithmetic or calculus operations. Sentences typically begin with imperatives like "Substitute the values into the formula," "Simplify the numerator and the denominator," "Calculate the numerator," "Integrate both sides," or "Solve for . . . ," and they perform the concrete manipulation needed to advance the solution. It does not include planning or outlining next steps, stating assumptions, recalling definitions, or presenting hypotheses – only the hands-on calculation moves. | "Substitute the values into the equilibrium expression:" |
| **Invoking Known Formulas and Laws** | These sentences serve to retrieve and state established mathematical or physical relationships from memory as ready-to-use premises. They include explicit references to canonical laws (e.g. "Gauss's Law states that. . . ," "Raoult's Law states that. . . "), definitions (e.g. "The de Broglie wavelength $\lambda$ is given by. . . "), and basic deductions grounded in scenario assumptions (e.g. "Since the box is on a horizontal surface, N = mg"). They do not introduce new hypotheses, express uncertainty, plan future steps, or provide high-level strategy; they simply supply the precise equations and principles needed for subsequent calculation. | "The heat required to vaporize a substance is given by the product of the number of moles, the molar heat of vaporization, and the temperature difference between the boiling point and the initial temperature." |
| **Asserting Propositions** | These sentences function to state domain-specific facts, rules, or conclusions as explicit propositions in the reasoning chain. They include the invocation of known principles (for example, economic or legal doctrines), the declaration of contract terms or definitions, and the direct application of those principles to arrive at intermediate or final conclusions (often signaled by words like "therefore," "thus," or "will"). What is included are both general normative or causal rules ("Reserve requirements decrease the money supply") and case-specific outcome statements ("The niece will probably win the in personam action against the farmer"). This category does not include sentences that propose new assumptions, plan future steps, express uncertainty, or generate and test hypotheses. | "<answer> The decision will be in favor of the uncle, based on the fact that the personal assistant has no valid claim for an accounting of the value of the oil removed and an injunction against further oil removal, as the conveyance was made with a condition that the personal assistant would receive the property if the uncle's wife died without issue." |
| **Answer Declaration** | These sentences serve to present the final result of a reasoning chain, signaling that all necessary steps have been completed and the solution is now being stated. They almost always begin with a conclusion marker such as "Therefore," "Thus," or "So," followed by phrasing like "the answer is," "the correct answer is," or "the final answer is," often with a boxed or standalone value. Included are any statements that explicitly announce the computed or deduced outcome. Excluded are intermediate calculations, explanatory comments, hypothesis proposals, or any steps that do not directly state the final solution. | "Therefore, the correct statement is $\boxed{B}$." |

| Category | Description | Top-Activating Example |
|---|---|---|
| **Arithmetic calculation steps** | These sentences each carry out a concrete numeric operation – multiplication, division, addition, subtraction, exponentiation, or unit conversion – presenting the expression and its evaluated result (for example, "11,180 $\times$ 0.03 = 335.40," "22,990,000 grams = 22,990 kg," "cos(2) $\approx$ –0.4161"). They serve purely to advance the computation by producing intermediate or final numerical values. This category does not include higher-level reasoning moves such as forming hypotheses, acknowledging uncertainty, planning next steps, or introducing conceptual definitions. | "- Speed: 7 mph = 7 $\times$ 1.60934 km/h = 11.26538 km/h = 11.26538 $\times$ 1000 m/h = 11265.38 m/h = 11265.38 / 3600 m/s $\approx$ 3.1293 m/s" |
| **Retrieving Constants and Formulas** | These sentences serve to pull in or restate standard numeric values (like R = 0.08206 L·atm/(mol·K), $\kappa$_air = 0.026 W/m·K), problem-given quantities (m_water = 185 g, $\Delta$H_vap = 9.7 kcal/mol), and general mathematical or physical relationships (Weight = Density$\times$Volume$\times$g, Volume = length$\times$width$\times$height). They establish the concrete premises and definitions that subsequent algebraic or numerical steps rely on. This category does not include speculative reasoning, planning of next steps, uncertainty acknowledgments, or actual manipulative inferences beyond stating those constants and equations. | "The specific gas constants for helium and nitrogen are approximately $R_{\text{He}} = 207.97$ ft$^3$ lb/lb · R and $R_{\text{N}_2} = 166.87$ ft$^3$ lb/lb · R, respectively." |
| **Subproblem Specification** | These sentences serve to articulate individual sub-tasks or questions within a broader reasoning process, effectively functioning as headings or prompts for the next step. They typically appear in the imperative ("Determine the Reynolds number:", "Identify the goddess:", "Calculate the molar solubility of CaF$_2$:") or as labeled scenarios ("Scenario 1: ...", "Option C: ..."), signaling what the reasoner will address next. This category does not include actual analytic content or domain knowledge statements, hypotheses, uncertainty acknowledgments, or detailed solution steps – only the declaration of the task to be undertaken. | "Reptilian Excretory System (Terrestrial):" |
| **Functional Explanation** | These sentences articulate the function, role, or causal effect of a concept, theory, behavior, or mechanism. They typically begin with "This..." or "It..." (or name the subject) and describe what that subject "leads to," "enables," "involves," "includes," or "provides." Included are statements that explain how something operates, why it matters, or what consequences it produces. Not included are speculative guesses, uncertainty acknowledgments, planning steps, or formal definitions of mathematical terms. | "Social conditioning: The society may use various forms of social conditioning, such as propaganda, advertising, or cultural norms, to shape individuals' perceptions and values, making it challenging for them to recognize their true nature." |
| **Recording Mathematical Expressions** | This category encompasses the bare presentation of equations, formulas, and algebraic or calculus-derived expressions as intermediate results in a solution. Included are any standalone mathematical statements – derivatives, integrals, rearranged formulas, numerical simplifications, partial-fraction decompositions, modular congruences, and substitutions – that advance the computation without accompanying commentary or strategic framing. Not included are sentences that express uncertainty, outline next steps, state assumptions in words, or offer high-level explanations or hypotheses. | $\frac{1}{s(s^2+4)} = \frac{1/4}{s} + \frac{-1/4s}{s^2+4} = \frac{1}{4s} - \frac{s}{4(s^2+4)}$ |

## G.3. Open-Reasoner-Zero-7B (Layer 20, Dict Size 10)

*Table 8.* Reasoning taxonomy for Open-Reasoner-Zero-7B (Layer 20, Dict Size 10). Categories are identified by our SAE; examples are top-activating sentences.

| Category | Description | Top-Activating Example |
|---|---|---|
| **Stating Background Knowledge** | These sentences supply domain-specific facts or definitions drawn from memory to underpin the reasoning. They include clear declarative statements of biological, chemical, anatomical, or physical properties (for example, "Peptidoglycan is a key component of the bacterial cell wall," "Vitamin K is essential for the synthesis of certain coagulation factors," or "In mitochondria, the proton pump is located in the inner membrane"). What is included are pure factual assertions or textbook-style definitions, without any indication of planning next steps, expressing uncertainty, forming hypotheses, or drawing new inferences. This category does not include sentences that pose questions, make assumptions, outline procedures, or propose explanations – it is limited to recalling and stating established facts. | "Collagenase is an enzyme produced by S. aureus that degrades collagen, a major component of connective tissue." |
| **Outline Headings** | These sentences function as organizational markers that label, segment, and structure the reasoning into discrete sections or steps. They include bulletpoints, numbered or titled steps (e.g., "### Step 2:...," "**Criteria for Seeking Professional Help:**"), and topic headings (e.g., "### Relationship Between Punishment and Extinction:"). They do not convey substantive analytical content, calculations, or argumentation but instead signal the next subtask or thematic section in the reasoning flow. | "- **War and Security Studies**:" |
| **Performing Calculation Steps** | This category captures sentences that enact specific mathematical or computational operations within a solution, signaling and carrying out concrete arithmetic, algebraic, or calculus procedures. Included are directives and enacted steps like "Simplify the expression:," "Perform the division:," "Calculate the value:," "Substituting the given values:," or "Taking the partial derivative...," often prefaced with "Now" or "Let's." These sentences do not plan overall strategy, introduce assumptions, recall definitions, or express uncertainty – they simply execute individual calculation moves in the reasoning. | "Now, perform the multiplication:" |
| **Calculation Steps** | These sentences carry out direct numerical or algebraic computations by substituting values into formulas, performing arithmetic operations (addition, multiplication, division, exponentiation), and simplifying results to concrete numerical expressions (often with units). They include intermediate evaluations, unit conversions, approximations, and the derivation of numeric quantities. They do not introduce new assumptions, propose or test hypotheses, outline future steps, or express uncertainty – instead, they focus exclusively on executing the arithmetic or algebraic work of the solution. | $\Delta V = \frac{-V_0 \Delta P}{K} = -\frac{2500\ \text{cm}^3 \times 1.519875 \times 10^7\ \text{dynes/cm}^2}{2 \times 10^{12}\ \text{dynes/cm}^2}$ |
| **Listing Given Data** | These sentences extract and record the explicit facts or numerical values provided by the problem statement. They consist of standalone declarations of parameters – masses, costs, concentrations, pressures, dimensions, rates, and other input values – often introduced with bullets or commas. This category does not include computations, hypothesis generation, uncertainty acknowledgments, or conceptual definitions; it is strictly the step of gathering and restating the raw inputs needed for subsequent reasoning. | "The cash price of the motorcycle is $275." |

*Continued from previous page*

| Category | Description | Top-Activating Example |
|---|---|---|
| **Recalling Standard Equations** | These sentences serve to retrieve and state well-known mathematical or physical relationships as premises for later steps. They typically use phrasing such as "is given by," "can be calculated using," "is defined as," or "the change in X is related to ..." to introduce a formula. Included are any generic, canonical equations or definitions (e.g., heat transfer laws, ideal-gas relations, work and energy formulas) that will be applied subsequently. Not included are speculative hypotheses, planning steps, declarations of uncertainty, or context-specific numerical computations. | "The heat transfer rate can also be expressed using the heat transfer coefficient $U$, the surface area $A$, and the temperature difference $\Delta T$:" |
| **Conclusion Statements** | These sentences explicitly draw the outcome of a line of reasoning or calculation, typically signaled by words such as "Therefore," "Thus," "So," or "In summary." They present the final result – often a numerical value or categorical determination – that follows from earlier steps without introducing new premises, exploring uncertainty, or outlining further actions. What is included are statements that synthesize prior derivations into a clear end point. What is not included are the intermediate computational steps, expressions of doubt, plans for future steps, or newly stated assumptions. | "Therefore, the percentage of the man's daily energy requirement that is met by protein is:" |
| **Issue Spotting** | These sentences perform the cognitive task of identifying and articulating discrete legal issues, claims, defenses, or elements by applying legal rules to specific fact patterns. They typically introduce questions ("The key question is whether ..."), label potential causes of action or defenses ("Negligence: ...", "Breach of Contract: ..."), and draw provisional inferences about outcomes ("If X can be established, then Y may succeed"). This category excludes sentences that merely state procedural plans, express uncertainty, set up assumptions, recall definitions, or outline high-level strategy – its focus is on mid-level application of law to facts. | "If the assignment was valid and the city now holds the rights to enforce the covenants, the city can pursue the chef for damages." |
| **Stepwise Analysis** | These sentences serve to segment and structure the reasoning by announcing the next analytical operation – breaking the problem into subparts, outlining which elements to examine, and guiding the flow of the solution. They typically begin with "Let's...," "Given these... let's...," or "Now, let's...," followed by verbs like break down, analyze, consider, examine, explore, or determine. What's included are signals that frame and introduce upcoming reasoning steps without yet performing the domain-specific deductions, computations, or evaluations themselves. What's not included are actual contentful hypotheses, uncertainty acknowledgments, formal assumptions, or retrieval of definitions – they merely mark the transition into detailed analysis. | "Let's break down the question and explore Nagel's views on skeptical arguments:" |

*Continued on next page*

| Category | Description | Top-Activating Example |
|---|---|---|
| **Conceptual Elaboration** | These sentences supply detailed explanations of concepts, mechanisms, or causal relationships that serve as background knowledge within the reasoning. They typically state domain facts or general properties using phrases like "This can lead to...," "This includes...," "These programs...," or "The concept of...," thereby enriching the context for subsequent inference. What's included are factual descriptions of how things work, definitions, and cause-and-effect statements presented as given. What's not included are planning directives ("first...next..."), expressions of uncertainty ("I'm not sure..."), speculative hypotheses ("maybe...could be..."), or explicit assumption statements ("let's assume..."). | "When individuals prioritize their own interests, it can lead to a breakdown in social cohesion and a lack of shared values and goals." |

### G.4. DeepSeek-R1-Distill-Qwen-1.5B (Layer 4, Dict Size 15)

*Table 9.* Reasoning taxonomy for DeepSeek-R1-Distill-Qwen-1.5B (Layer 4, Dict Size 15). Categories are identified by our SAE; examples are top-activating sentences.

| Category | Description | Top-Activating Example |
|---|---|---|
| **Identifying Additional Factors** | This cluster consists of statements in which the reasoner broadens the analytical scope by calling out extra variables, contexts, or perspectives that might influence the problem. They typically begin with prompts like "I should also consider...," "I also think about...," "What about...?," or "Another thought:," and name a potential factor (e.g., historical context, legal process, psychological effects). These are not planning steps (no "first/next/then"), not explicit assumptions, nor expressions of uncertainty, but rather a brainstorming of relevant considerations to incorporate into the ongoing analysis. | "I also think about the ethical implications." |
| **Algebraic Transformation** | This category comprises sentences that perform direct symbolic or algebraic manipulations on equations – rearranging terms, isolating variables, dividing or factoring expressions, and applying mathematical identities. These steps often begin with transitional cues such as "So," "Dividing both sides by," or similar phrases, and then present the newly obtained formula. What's included are the mechanical computation steps that move from one algebraic form to another (e.g. solving for $\Delta V$, expressing $\lambda/2$, isolating $C^3$). Not included are speculative comments, uncertainty acknowledgments, planning directives, or high-level definitions; these sentences strictly enact immediate equation rewriting. | "$T_0 - T = (1/2) V^2 + (\gamma - 1) V^2 / (2M)$" |
| **Acknowledging Uncertainty** | These sentences serve to flag gaps in knowledge or confidence by directly stating doubt or incomplete information, typically with phrases like "I'm not sure," "I don't know," "I'm not entirely certain," or "but I'm not entirely sure." They function to pause the reasoning flow and signal that a point may need further evidence, clarification, or verification. This category includes straightforward expressions of not knowing or being unsure, without immediately offering a hypothesis, plan, or new assumption. It does not include sentences that introduce tentative explanations ("perhaps...," "it could be that..."), outline next steps in the reasoning, or assert working assumptions. | "But I'm not entirely sure if I'm missing something." |

| Category | Description | Top-Activating Example |
|---|---|---|
| **Reevaluation Signals** | These are brief interruptions in the reasoning where the thinker pauses to question, doubt, or challenge a just-made claim or assumption and prompts reanalysis. They typically begin with "Wait, but…," "But wait…," or "hold on" and introduce a specific objection or point needing clarification. This category does not include general expressions of uncertainty ("I'm not sure") or plans for future steps, but is narrowly about signaling an immediate re-examination of the current line of thought. | "Wait, but wait." |
| **Problem Restatement** | These sentences perform the initial articulation of the task by paraphrasing or summarizing exactly what question needs to be answered. They typically begin with phrases like "Okay, so I need to figure out…," "I have this problem…," or "First, I need to determine…," and they identify the goal or end-state (e.g., finding a percentage, locating the right law, computing a physical quantity). Included are any statements that frame "what am I being asked to do?" without yet executing computations, introducing technical definitions, or evaluating uncertainty. Not included are sentences that propose tentative explanations (hypotheses), recall formulas, state assumptions, express doubt, or carry out actual solution steps. | "Okay, so I'm trying to figure out this question about a 31-year-old woman with type 2 diabetes who has a 4-cm necrotizing wound on her foot." |
| **Invoking Known Equations** | These sentences retrieve and state established mathematical or physical relations from memory – canonical formulas, definitions, and laws – without yet applying them in detailed calculation. They often begin with "I remember that," "I think the formula is," "By definition," or "The law states," followed by a named principle or equation (e.g., $PV = nRT$, Stefan–Boltzmann law, Newton's law of cooling). What's included are explicit recalls of domain knowledge in the form of equations or definitions. What's not included are planning statements that outline future steps, expressions of uncertainty, or tentative hypotheses and assumptions. | "The heat released by the air as it cools down is $Q = m\,c\,\Delta T$, where m is the mass of the air, c is the specific heat capacity of air, and $\Delta T$ is the temperature change." |
| **Metacognitive Self-Prompts** | These are the speaker's internal "let me…" prompts used to manage and direct the reasoning process. They signal a forthcoming operation – recalling a fact ("Let me try to recall"), converting or computing ("Let me convert…", "Let me calculate that"), verifying or breaking down steps ("Let me double-check", "Let me break this down step by step") – without yet providing substantive domain content. This category includes only these procedural cue phrases and excludes the actual content of facts recalled, calculations performed, hypotheses generated, or conclusions drawn. | "Let me think." |
| **Proposing Possible Explanations** | These sentences advance the reasoning by offering tentative causal or functional explanations for phenomena, often tying one factor to another in a "what might cause what" fashion. They typically deploy modals ("might," "could," "may") or framing cues ("For example," "This could lead to," "Maybe") to introduce hypothetical links or scenarios. Included are speculative "might be," "could lead to," or "for instance" statements that put forward coherent possibilities for further consideration. Excluded are mere expressions of doubt without an explanatory proposal, explicit planning steps, formal assumptions, or recalls of established definitions. | "When people talk to others about their eating habits, they might learn healthier ways to consume their food, which can contribute to a more balanced diet and overall health." |

| Category | Description | Top-Activating Example |
|---|---|---|
| **Articulating Intermediate Legal Conclusions** | These sentences serve to state the immediate argumentative conclusion or position of a party based on the facts just discussed. They typically use inference markers ("So," "Therefore," "But," "However," "Alternatively") and frame propositions as one side's claim or defense ("the store's defense is that. . . ," "the customer's claim is that. . . ," "the defendant's defense is. . . "). Included are any statements that draw an interim legal inference – summarizing what must follow given the premises or what a party will argue in court. Not included are sentences that lay out next steps in reasoning, express doubt or uncertainty, introduce assumptions, or recall formal definitions. | "But the owner is bringing a claim against the newspaper, so the newspaper is arguing that the owner's actions were unreasonable because the editor refused to publish the ad." |
| **Validating reasoning steps** | These sentences serve as self-checks that confirm or question the correctness and plausibility of an intermediate result or logical step in the reasoning process. They include affirmative validations ("That seems right," "That seems correct"), expressions of doubt ("Hmm, that can't be right," "But that doesn't make sense"), and direct queries ("Is that right?"), all aimed at ensuring consistency before proceeding. This category does not cover proposing new ideas, planning next moves, stating assumptions, or recalling definitions – its sole focus is on evaluating whether a just-completed step holds up. | "That seems right." |
| **Recalling Background Knowledge** | These sentences serve to retrieve and state previously learned facts, definitions, or domain concepts that the reasoner brings to mind to support ongoing problem-solving. They often begin with framing phrases like "I remember that. . . ," "I think. . . ," or simple declarative statements of factual content (e.g. "cyanobacteria are photosynthetic," "chylomicrons are secreted by enterocytes"). What is included are pure memory-recall moves – unstaged statements of known information without immediate inference, calculation, or planning. What is not included are tentative hypotheses about causes, explicit plans of next steps, statements of uncertainty about the reasoning process, or on-the-fly derivations and computations. | "But wait, bacteria have a cell wall made of peptidoglycan, which is a carbohydrate, while plant cell walls are made of cellulose, which is also a carbohydrate." |
| **Suggesting Possibilities** | These sentences introduce concrete alternative explanations or interpretations for unresolved aspects of the problem, using speculative cues like "maybe," "perhaps," "or perhaps," and "alternatively." They serve to generate hypotheses about which variable, term, option, or concept might fit when the correct choice is unclear. Included are tentative proposals that guide the reasoning toward testing or elimination ("Maybe it's a different variable?" "Perhaps the question refers to the legal term."). Excluded are mere expressions of uncertainty without offering options, explicit planning steps, recalled definitions, or firm assumptions. | "Maybe it's B or something else." |
| **Numeric Computation** | These sentences execute individual arithmetic and approximation steps needed to advance the solution, such as multiplying coefficients, dividing quantities, adding partial products, rounding intermediate results, and converting units. They present explicit numeric evaluations (often with "$\approx$" or "about") that feed directly into the larger calculation. This category includes only the act of computing or simplifying numbers – not the articulation of strategy, the invocation of formulas, the statement of assumptions, or the expression of uncertainty. | "Then, 126,720 0.000086667 $\approx$ 126,720 0.00008 = 10.1376" |

| Category | Description | Top-Activating Example |
|---|---|---|
| **Summarizing Problem Details** | These sentences serve to restate or paraphrase the facts and questions as presented in the prompt, ensuring the reasoner has correctly captured all given information before proceeding. They include characterizations of patients (e.g. "The patient is a 25-year-old man…"), descriptions of scenarios or contracts ("Carol bought a 1964 Thunderbird…"), and explicit reformulations of what the question is asking ("The question is asking what the expected physical finding would be next."). They do not contain calculations, hypothesis generation, or argumentative moves, but purely re-express the problem's inputs and objectives. | "A 25-year-old man is coming in with a 6-day history of fever, severe muscle pain, and diffuse, painful swelling in his neck, underarms, and groin area." |
| **Stating intermediate results** | These sentences explicitly present the direct outcomes or conclusions drawn from the immediately preceding reasoning steps, typically introduced by discourse markers like "So," or "Therefore." They include declarations of derived formulas, assigned values, simplified expressions, or assertions that a condition holds. This category does not include planning or speculative statements, expressions of uncertainty, definitions or recalled facts, nor the introduction of new assumptions – only the clear announcement of what has just been deduced. | "So, the answer is volume." |

## G.5. DeepSeek-R1-Distill-Llama-8B (Layer 6, Dict Size 15)

*Table 10.* Reasoning taxonomy for DeepSeek-R1-Distill-Llama-8B (Layer 6, Dict Size 15). Categories are identified by our SAE; examples are top-activating sentences.

| Category | Description | Top-Activating Example |
|---|---|---|
| **Recalling Relevant Formulas** | These sentences retrieve and state specific mathematical or physical relationships from memory to serve as building blocks for later calculation or analysis. Each one presents a canonical equation or law – often introduced by cues like "The formula is," "Recall that," or explicit notation of variables and constants – without yet applying it to compute a result. This category includes standalone declarations of standard formulas (e.g., heat transfer, momentum, ideal gas law) and not procedural steps, interpretations of data, or speculative reasoning about what might happen next. | "So, the formula I need is $Q = m\,c\,\Delta T$, where $Q$ is the heat energy, $m$ is mass, $c$ is specific heat capacity, and $\Delta T$ is the temperature change." |
| **Retrieving Background Knowledge** | These sentences bring up established facts, definitions, or domain concepts from memory to ground the ongoing reasoning. They often begin with cues like "I remember," "From what I recall," "I think," or "as I understand," followed by a concise statement of a known element (e.g., the structure of the ETC, the meaning of a z-score, the branches of government). This category does not include planning future steps, expressing mere uncertainty, proposing new hypotheses, or setting fresh working assumptions. | "Glands are like little organs that make hormones, right?" |

*Continued from previous page*

| Category | Description | Top-Activating Example |
|---|---|---|
| **Listing Additional Considerations** | This category captures moments when the reasoner expands the scope of their analysis by appending new factors, dimensions, or alternative methods to examine. Included are sentences introduced with phrases like "I also think about…", "I also wonder about…", "I should also consider…", "Another thought…", or "Alternatively…", each adding a fresh angle without yet evaluating or ruling it out. These statements are neither firm assumptions nor concrete plans but serve to broaden the set of points under review. Not included are expressions of uncertainty about a single claim, explicit tentative hypotheses, step-by-step plans, or definitions and recollections of known facts. | "I also think about the situation itself." |
| **Articulating Conditional Legal Hypotheses** | These sentences frame "if–then" style possibilities to explore how varying facts or interpretations could alter legal outcomes. They typically begin with conditionals ("If," "But if," "Alternatively") and employ modal verbs ("might," "could," "would," "should") to signal speculative legal arguments or defenses. Included are tentative statements about liability, contract validity, or statutory application under different scenarios. Excluded are definitive rule recitals, step-by-step procedural plans, or pure factual assertions without hypothetical qualification. | "So, maybe the landlord can't claim breach of lease because the fire wasn't their fault." |
| **Conditional Causal Reasoning** | This cluster captures steps where the reasoner applies "if–then," "when," "because," or "so" constructions to spell out how a change in one variable or action produces an effect in another. Included are sentences that directly propagate causal or logical consequences ("If the money supply increases, people have more to spend," "When the Fed buys bonds, interest rates fall," "Because temperature changes, kinetic energy shifts"). Not included are statements of uncertainty, planning of next steps, explicit assumption setting, or the recall of formal definitions. | "So, if the money supply increases, people have more money to spend, which shifts the aggregate demand curve to the right, not the supply." |
| **Drawing Conclusions** | These sentences synthesize prior reasoning steps and evidence into a definitive answer or judgment, typically signaled by discourse markers like "Therefore," "So, putting it all together," "In summary," or "In conclusion." They explicitly integrate multiple strands of analysis into a cohesive final claim or solution. This category includes only explicit concluding statements that present the outcome of the reasoning. It does not include hypothesis generation, expressions of uncertainty, planning steps, assumption declarations, or recall of definitions. | "So, putting it all together, the correct answer is C) Mutation-based fuzzing is a type of black-box testing." |
| **Summarizing Given Data** | These sentences serve the function of collecting and restating the explicit parameters, measurements, and conditions provided by the problem. They typically begin with numerical values and units (e.g., "The tube is 19 feet long…," "The air flows at 3.0 lbm/hr," "A 50 mole % solution…"), or otherwise recast the scenario's raw inputs. Included are any lines that purely enumerate or paraphrase the problem's known facts without analysis, inference, planning, or hypothesis. Not included are sentences that introduce new assumptions, outline next steps, recall definitions, express uncertainty, or propose explanations. | "The plane weighs 18,000 lbf, and the normal speed in level flight is 230 ft/sec when the atmospheric density is 0.076 lbm/ft$^3$." |

*Continued from previous page*

| Category | Description | Top-Activating Example |
|---|---|---|
| **Self-Checking** | These sentences are the model's metacognitive pauses to question or verify the accuracy of a prior step or calculation. They typically begin with markers like "Wait," "Hold on," or "Let me double-check," and include phrases such as "is that correct?", "maybe I made a mistake," or "now I'm confused." This category is distinct from proposing new hypotheses, planning future reasoning steps, or stating assumptions – its sole function is to reassess and confirm the correctness of what was just said. | "Wait, hold on, is that correct?" |
| **Result Confirmation** | These sentences serve to acknowledge and affirm the correctness or consistency of a just-completed inference, calculation, or deduction. They typically use brief evaluative phrases such as "That seems correct," "That seems right," or "Yes, that's correct" immediately after an intermediate step. Included are confirmations that a formula, identity, or logical conclusion aligns with expectations. Excluded are statements that introduce new hypotheses, express uncertainty, set up future steps, or state assumptions. | "That seems right." |
| **Calculation step** | These sentences carry out concrete mathematical operations – defining or assigning expressions, differentiating or integrating formulas, simplifying or rearranging equations, and computing specific variable values. They typically feature an equals sign or a differential operator (e.g., "h′(x)=...," "C′(r)=d/dr[...]," "So, T=...," "Then $y_{n+1}=y_n+...$"). Included are algebraic and calculus manipulations that transform previously stated formulas into new forms. Excluded are high-level planning remarks, speculative hypotheses, uncertainty acknowledgments, assumption statements, or pure definitions unaccompanied by an immediate calculation. | "$P_t = (M_0/T_0)V(1+g_M)^t/(1+g_T)^t$" |
| **Arithmetic Computation** | This category covers individual steps where the reasoner carries out explicit numeric operations – multiplications, divisions, additions, subtractions, exponent manipulations, unit conversions, and rounding – to produce intermediate values. Included are approximate "mental math" calculations (e.g., breaking a product into parts, adjusting for small differences, converting units or magnitudes, and computing powers or roots). Not included are higher-level moves such as stating assumptions, planning future steps, recalling definitions, proposing hypotheses, or discussing uncertainty – this category is strictly about computing numbers. | "Then, $0.1642 \cdot 273.15 \approx 0.1642 \cdot 273 \approx 44.8$ L" |
| **Restating the Question** | These sentences serve to frame the problem by explicitly articulating the task or question the reasoner intends to address next. Each statement summarizes or clarifies what needs to be determined – often using phrasing like "I'm trying to figure out..." or "The question is asking...." Included are all utterances that recast the original prompt in the reasoner's own words to set the goal of the solution. Not included are steps that recall specific definitions, make assumptions, propose hypotheses, plan detailed solution steps, or perform actual calculations. | "Okay, so I'm trying to figure out what happens when an attitude is communicated." |

*Continued on next page*

*Continued from previous page*

| Category | Description | Top-Activating Example |
|---|---|---|
| **Generating Hypothetical Explanations** | These sentences introduce tentative causal links or possible mechanisms, using qualifiers such as "could," "might," or "maybe" to offer one or more plausible ways that an effect could arise or a process could work. They sketch out speculative scenarios (e.g., "This could lead to...," "Perhaps through...," "Maybe the invention of...") without asserting any as definitive. Included are sentences that explore potential outcomes, intermediary steps, or underlying factors. Excluded are statements of definite fact, expressions of pure uncertainty without a proposed mechanism, planning steps, formal definitions, or purely declarative observations. | "They might look for new ways to belong or find meaning, which could lead to things like consumerism or joining other groups, maybe even cults or gangs, as a way to feel connected." |
| **Procedural Step Planning** | These sentences explicitly frame the upcoming operations in the reasoning process, using cues like "Let me...," "First," "Next," or "step by step" to outline and initiate each subtask. They include announcing intentions to write down equations, define variables, compute parts of an expression, or break down a problem into smaller pieces before executing calculations. They serve as organizational scaffolding rather than introducing new assumptions, definitions, or hypotheses. This category excludes recall of facts or expressions of uncertainty – it purely signals the next procedural move. | "Let me write that out step by step." |
| **Expressing Uncertainty** | These sentences explicitly acknowledge gaps or doubts in the reasoner's knowledge by using hedges and disclaimers such as "I'm not sure," "not 100% certain," "I might need to check," or "I should double-check." They signal awareness of limitations or ambiguity in the current line of thought without introducing new hypotheses, committing to assumptions, or outlining concrete next steps. This category does not include sentences that propose a specific explanation to be tested, state explicit assumptions, or plan future operations. | "I think it's the latter, but I'm not entirely sure." |

## G.6. DeepSeek-R1-Distill-Qwen-14B (Layer 38, Dict Size 5)

*Table 11.* Reasoning taxonomy for DeepSeek-R1-Distill-Qwen-14B (Layer 38, Dict Size 5). Categories are identified by our SAE; examples are top-activating sentences.

| Category | Description | Top-Activating Example |
|---|---|---|
| **Numeric Computation** | These sentences execute concrete arithmetic or algebraic manipulations – multiplying, dividing, adding, subtracting, taking roots or powers, converting units, factoring, and plugging numerical values into formulas – to produce intermediate or final numerical results. They often break down complex operations into smaller mental-math steps (e.g., "$4210 \times 9.36 \approx 4210 \times 9 + 4210 \times 0.36$"). They do not introduce new hypotheses, express uncertainty, outline future steps, state assumptions, or recall conceptual definitions. | "Then, 26.562e-12 3.570 $\approx$ 26.562 3.570 * 1e-12" |

*Continued from previous page*

| Category | Description | Top-Activating Example |
|---|---|---|
| **Conditional Reasoning** | These sentences generate and explore "if–then" scenarios and alternative possibilities, using modals ("might," "could"), framing phrases ("Maybe," "Alternatively"), or explicit conditionals ("If…then…") to propose and evaluate hypothetical outcomes. They often introduce speculative links or exceptions (e.g., "One possible exception is…," "Perhaps the duty…") and then assess their plausibility in context. This category does not include merely stating unknowns without elaboration (pure uncertainty), laying out fixed assumptions for calculation, or planning concrete next steps; instead it focuses on formulating and testing conditional hypotheses. | "So, the city's defense that it's not a commerce issue might be weak because the court could find that it unduly burdens interstate commerce, even if the intent was public safety." |
| **Problem Restatement** | These sentences rephrase and define the task or question to be solved, specifying what is being asked and summarizing the given scenario or data. They often start with phrases like "Okay, so I have this problem…" or "First, I need to identify the given values," and serve to orient the solver by clarifying the goal. This category does not include sentences that lay out detailed solution plans, recall formal definitions, propose assumptions, or perform actual calculations. | "First, I need to list all the plant heights from the given data: 161, 183, 177, 157, 181, 176, 180, 162, 163, 174, 179, 169, 187." |
| **Metacognitive Markers** | These are brief self-addressed cues that regulate the flow of thinking by pausing, refocusing, or preparing the next move in the reasoning process. They include hesitations and transitions using phrases like "Let me think," "Hmm," "Wait," or "Let me break this down step by step." This category covers utterances that manage or signal shifts in thought rather than carrying substantive content, such as definitions, data calculations, hypothesis proposals, or explicit multi-step plans. | "Hmm, I'm a bit rusty on this, but let me try to think it through." |
| **Recall of Domain Knowledge** | These sentences function to retrieve and state established facts, definitions, laws, or background information from memory as foundational premises for further reasoning. They often begin with framing phrases such as "I remember that," "I know that," "From what I recall," or directly present canonical statements (e.g., "Raoult's Law states…"). What is included are declarative recitations of known concepts, formulas, or classifications. What is not included are speculative hypotheses, uncertainty acknowledgments, planning steps, or newly derived conclusions. | "There's plankton, which I think are the tiny plants and animals that float in the water and can't move on their own." |

## G.7. DeepSeek-R1-Distill-Qwen-32B (Layer 27, Dict Size 15)

*Table 12.* Reasoning taxonomy for DeepSeek-R1-Distill-Qwen-32B (Layer 27, Dict Size 15). Categories are identified by our SAE; examples are top-activating sentences.

| Category | Description | Top-Activating Example |
|---|---|---|
| **Proposing Explanations** | These sentences advance tentative causal or interpretive ideas about the topic at hand, often signaled by modals ("might," "could," "maybe," "probably") or framing phrases ("this could mean," "another common misconception is," "it probably argues"). They introduce one or more possible reasons, mechanisms, or implications to account for observations or to make sense of concepts. This category does not include pure expressions of uncertainty without content (e.g., "I'm not sure"), explicit planning statements (e.g., "Next, I will. . . "), formal assumption declarations (e.g., "Assume that. . . "), or the straightforward recall of definitions. | "Maybe also mention how emotions can help in understanding others' perspectives, fostering moral growth, and building social connections, which are essential for a moral society." |
| **Enumerating additional factors** | These sentences serve to broaden the scope of the analysis by prompting the reasoner to add new angles, contexts, or variables for consideration – often signaled by phrases such as "I should also consider. . . ," "I should also think about. . . ," "Wait, maybe. . . ," or "Alternatively. . . ." They explicitly list extra domains (e.g. historical, political, social, environmental, methodological) and potential complications or exceptions that might affect the solution. They do not lay out a precise sequence of steps, state formal assumptions, recall definitions, or propose a single testable hypothesis; rather, they simply remind the reasoner to include more relevant factors in their thinking. | "I should also consider the impact on inflation." |
| **Problem Framing** | These sentences mark the very start of the reasoning episode by restating or paraphrasing the question and by recalling or naming relevant background concepts. They typically open with phrases like "Okay, so I need to figure out. . . ," "First, I remember that. . . ," or "Let me start by. . . ," thereby setting up what must be found or understood before any actual solution steps are carried out. Included here are all instances where the model defines the task, breaks it down into sub-questions, or summons prior knowledge to frame its approach. Excluded are sentences that begin to execute computations, propose hypotheses, evaluate uncertainty, or state assumptions about the solution. | "Okay, so I need to figure out what directly determines a competitive firm's demand for labor." |
| **Inferring Causal Effects** | These sentences explicitly link a change in one variable or condition to its logical or physical consequence, using markers like "because," "so," "implies," "would," or "might." They spell out how one premise (e.g., a shift in supply, a change in interest rates, or an alteration in cost) produces a particular outcome (e.g., price movement, production adjustment, or revenue change). Included are clear cause-and-effect explanations in domains such as economics, physics, and mathematics. Not included are mere plans of action, definitions or recalls of concepts, purely tentative guesses without causal linkage, or expressions of uncertainty without an explicit cause-to-effect chain. | "That makes sense because if it's more expensive for banks to borrow, they'll borrow less, which in turn reduces the amount of money they can lend out." |

| Category | Description | Top-Activating Example |
|---|---|---|
| **Computational step initiation** | These sentences signal the start of a concrete arithmetic or algebraic operation, breaking the problem into fine-grained "compute this" or "plug in that" actions. They include imperatives and planning phrases like "Let me compute that step by step," "Let me write that down," "Substituting back into...," or "Compute kinetic energy," where the LLM is about to perform or segment a calculation. This category does not cover higher-level strategy planning (e.g. "First outline the proof"), stating assumptions, recalling definitions, or expressing uncertainty – only the invocation of immediate computation or algebraic manipulation. | "Let me compute this step by step." |
| **Memory Retrieval** | These sentences serve to pull domain-specific facts, definitions, and relationships from long-term memory as premises for further reasoning. They often begin with "I remember," "I think," or "I've heard," and state known physiological, chemical, or general-knowledge details. This category includes explicit factual recalls used to ground subsequent steps. It does not include speculative hypotheses, uncertainty acknowledgments, planning statements, or formal mathematical definitions. | "The bulbourethral glands, also called Cowper's glands, I think they produce a clear fluid that acts as a lubricant." |
| **Verifying Intermediate Results** | These sentences are self-checks where the reasoner pauses to confirm or question the correctness of a just-completed calculation, inference, or statement. They include brief evaluative phrases like "That seems right," "Hmm, that doesn't seem correct," or "Wait, let me double-check," signaling validation or reconsideration of an intermediate result. This category captures neither planning future steps nor introducing new assumptions or hypotheses; it is strictly about monitoring and validating ongoing reasoning. | "Hmm, that seems correct." |
| **Drawing Conclusions** | These sentences serve to present the final result or resolution of the preceding line of thought. They typically signal that all prior analyses have been integrated – using cues such as "So, putting it all together," "Therefore," or "So, yeah, I think..." – and then state the answer or conclusion. This category includes any assertion of the outcome based on earlier reasoning steps. It does not include statements that introduce new hypotheses, express uncertainty, outline future steps, or recall definitions; its sole function is to close the loop by proclaiming the end-point of the reasoning. | "So, in conclusion, the overall order of the rate equation is 3." |
| **Stating Given Problem Data** | These sentences systematically recite the explicit numerical parameters and initial conditions drawn directly from the problem statement – masses, volumes, temperatures, pressures, dimensions, and other values that will feed into calculations. They typically use phrases like "given," "the problem states," or simple assignment notation (e.g., "$V = 3.00\,dm^3$") to capture each datum. This category includes only the act of identifying and recording the provided facts; it excludes any sentences that perform computations, outline solution steps, recall theoretical formulas, or express uncertainty or hypotheses. | "The given data is: mass of argon is 12.0 grams, initial volume is 1.0 $dm^3$ at 273.15 K, and it expands to 3.0 $dm^3$." |

*Continued from previous page*

| Category | Description | Top-Activating Example |
|---|---|---|
| **Evaluating Expressions** | These sentences carry out concrete computation steps – looking up values in tables or data structures, performing arithmetic or modular operations, simplifying algebraic sub-expressions, and evaluating logical or truth-table entries. Each instance records the immediate result of a small calculation (for example, "$3^4$ ... mod 7 is 4," "From the third row, third column ... = 0," "Second iteration: $3 - 2 = 1$," "Compute H: F = T"). This category does not include high-level moves such as proposing plans, stating assumptions, recalling broad definitions, or generating hypotheses; it is strictly about executing and reporting elementary evaluation steps. | "Then applying (1 2 3): 1 goes to 2, 2 goes to 3, 3 goes to 1." |
| **Intermediate Numerical Computations** | This cluster consists of individual steps where the reasoner carries out explicit arithmetic or unit-conversion operations – multiplying, dividing, converting units, handling exponents, and producing approximate numerical results. Included are sentences that show a numeric expression followed by its evaluated value (e.g., "0.0821 * 298.15 $\approx$ 24.465 L·atm/mol"), often with on-the-fly decimal approximations or conversions between units. These are not planning statements, assumptions, or definitions, but the concrete execution of basic calculation steps needed to advance the solution. | "Let's see, 1.38 * 273 is approximately 376.74, so 376.74e-23 J. So n = 101325 / 376.74e-23." |
| **Recalling Scientific Laws and Formulas** | These sentences retrieve established domain knowledge – laws, unit conversions, definitions, and canonical equations – from memory to serve as foundational premises in reasoning. They often begin with verbs or phrases like "I remember," "Recall that," or directly state "X's Law states" followed by a formal relationship or unit definition. This category includes pure declarations of known physical, mathematical, or chemical facts without applying them to specific problem data. It does not include planning steps, expressing uncertainty, proposing assumptions, or generating new hypotheses. | "It's defined as the ratio of stress to strain in the elastic limit." |
| **Recalling Formulas** | These sentences serve to pull standard mathematical or physical relationships directly from memory as ready-to-use premises. They consist of standalone expressions – such as $v_{nrms} = \sqrt{(3RT/M)}$, $\gamma = 1/\sqrt{(1-v^2/c^2)}$, $\mu(t)=e^{\int P(t)dt}$, or the Nernst equation – that the reasoner will later plug into a derivation or calculation. They do not propose new ideas, acknowledge uncertainty, plan next steps, or perform algebraic manipulations; instead, they simply state canonical formulas to be employed in subsequent reasoning. | "q_w = Q / A = (m_dot * (h_out - h_in)) / ($\pi$DL)" |
| **Expressing Uncertainty** | These sentences serve to signal the reasoner's awareness of gaps or potential errors in their current understanding. They use hedging language – "I'm not sure," "I think," "maybe," "I should double-check," "I'm a bit confused" – to qualify statements and indicate tentativeness rather than commitment. This category does not include concrete proposals for testing a specific hypothesis or detailed plans of action; instead, it purely acknowledges ambiguity or incomplete information. Definitive assertions, formal planning steps, and explicit hypothesis generation are excluded from this cluster. | "I think I need to confirm this, but since I can't look it up right now, I'll go with "moral particularism" as the answer, but I'm not 100% sure." |

*Continued from previous page*

| Category | Description | Top-Activating Example |
|---|---|---|
| **Stating Legal Rules** | This category consists of sentences in which the reasoner retrieves and articulates from memory the formal legal doctrines, statutory definitions, and rule-elements that will serve as the basis for analysis. Included are verdict-neutral statements of law – e.g. "In contract law, a unilateral mistake is when one party is mistaken...," "The plain view doctrine requires...," or "Malice can be express or implied..." – as well as recapitulations of test elements ("to prove malicious prosecution the plaintiff must show..."). What is not included are speculative "maybe" hypotheses without invoking a rule, forward-looking plans ("first I will..."), expressions of uncertainty ("I'm not sure if..."), or pure factual assumptions. | "In this case, the adjuster knew the pedestrian was entitled to compensation but told him he wasn't. That sounds like a knowing false statement, which could be fraud." |

### G.8. Open-Reasoner-Zero-32B (Layer 27, Dict Size 15)

*Table 13.* Reasoning taxonomy for Open-Reasoner-Zero-32B (Layer 27, Dict Size 15). Categories are identified by our SAE; examples are top-activating sentences.

| Category | Description | Top-Activating Example |
|---|---|---|
| **Factual Premise Retrieval** | These sentences function to retrieve and state concrete domain knowledge that serves as the factual foundation for subsequent reasoning. Included are declarative definitions ("Osmotic Pressure: ..."), anatomical or biochemical facts ("The carpal tunnel contains the median nerve..."), reaction equations, and process descriptions drawn directly from memory. Not included are planning statements, uncertainty acknowledgments, hypothesis proposals, or any sentences that perform calculations or draw inferences from these facts. | "2,3-bisphosphoglycerate $\rightarrow$ 3-phosphoglycerate (catalyzed by phosphoenolpyruvate carboxykinase and other reactions, but this is a simplification;" |
| **Specifying Variables and Constants** | These sentences enumerate the symbols, parameters, and numerical values (with units) that will be used in the calculation, linking each letter or symbol to its physical meaning or given magnitude. They include phrases like "where R is the ideal gas constant," "– g is the acceleration due to gravity," or "$R_1 = 0.60$ m as the radius of sphere 1." They do not perform calculations, set up logical deductions, introduce uncertainties, nor outline future steps – instead, they simply define the vocabulary and data for the ensuing reasoning. | "Here, $R$ is the ideal gas constant, $T$ is the temperature in Kelvin, and $M$ is the molar mass of the gas." |
| **Structuring Problem-Solving Steps** | These sentences function as labels for discrete, ordered actions in the reasoning process, using markers like "Step 3:", "Step 4:", or section headings (e.g. "### Step 4: Temperature conversion"). They explicitly announce what the next calculation or analysis phase will be, without yet performing it. Included are forward-looking planning statements that segment the solution into numbered or titled stages. Not included are sentences that execute computations, recall definitions, express uncertainty, or state assumptions. | "### Step 5: Properties of Copper" |

*Continued on next page*

| Category | Description | Top-Activating Example |
|---|---|---|
| **Deducing Quantitative Relationships** | These sentences present the immediate outcome of a reasoning sub-step by equating a target quantity to a specific expression or formula, often signaled by "So," "Therefore," or "Thus." Each one applies previously stated definitions, assumptions, or physical laws to derive a concrete algebraic or numerical relationship needed for the solution. Included are statements like "So, the total heat required … is:" or "Therefore, the work done … will be equal to … ." Not included are sentences that introduce new assumptions, outline future steps, recall abstract definitions without application, or express uncertainty. | "The amount of mercury that overflows will be the difference between the increase in volume of the mercury and the increase in volume of the glass flask:" |
| **Retrieving Background Knowledge** | These sentences recall and present established facts about thinkers, their works, dates, and conceptual contexts, supplying the factual groundwork for subsequent reasoning. They typically feature formulations like "X is known for…," "was published in…," or "One of the most famous is…," and name authorities, publications, and historical details. This category does not include speculative hypotheses, expressions of uncertainty or doubt, explicit planning steps, or formal mathematical definitions. | "Moore, a prominent philosopher in the early 20th century, who is well-known for his work in ethics and metaphysics." |
| **Recalling Legal Doctrines** | These sentences retrieve and state established legal rules, tests, and definitions as foundational premises for the analysis. They typically begin with phrases like "Generally," "In contract law," or label a doctrine (e.g. "Nexus Test:" "Best Evidence Rule:"), presenting canonical conditions and elements that apply to the facts. Included are clear declarations of legal standards (e.g. elements of misrepresentation, waiver, promissory estoppel, hearsay exceptions) used to ground subsequent reasoning. Excluded are speculative hypotheses, planning or procedural steps, personal uncertainty acknowledgments, or case-specific factual inferences. | "If the developer repudiates, the nephew is discharged from his duty to perform further under the contract with the developer and can treat the contract as ended." |
| **Evaluating Options** | This cluster represents the reasoning steps where the model weighs and judges candidate interpretations, explanations, or solution paths against the problem context. These sentences frequently begin with contrastive markers (e.g., "However," "But," "So," "Since") and use qualifiers ("seems," "might," "unlikely," "unless") to tentatively accept, reject, or refine possibilities. Included are judgments like "this seems less likely," "this might be misleading," or "this is true" when applied to a specific candidate. Not included are forward-planning statements, bare expressions of uncertainty without evaluating a specific option, explicit assumptions-setting, or recall of definitions. | "But this scenario is less likely given that it's specified as a "neon tube."" |
| **Numeric Calculation Steps** | These sentences perform explicit arithmetic operations needed in the solution, including multiplication, division, addition, subtraction, exponentiation, root extraction, fraction simplification, and unit conversions. They present concrete numeric evaluations – often using the "=" sign and occasionally "≈" for approximations – and compute intermediate or final values. This category does not include conceptual explanations, hypothesis generation, planning steps, uncertainty qualms, or recalling definitions; it is strictly the execution of numerical calculations. | Profit in USD $= 17.856 - 12.80435 = 5.05165$ USD |

*Continued from previous page*

| Category | Description | Top-Activating Example |
|---|---|---|
| **Generating Candidate Answers** | This category covers sentences in which the reasoner explicitly brainstorms or lists possible statements, choices, or answer options to address the question at hand. Such sentences typically begin with "Let's consider....," "Now, let's think about....," or "Given these considerations, let's....," and they open up alternative ways to phrase questions or potential answers. Included are purely generative moves that expand the solution space (e.g., proposing hypothetical statements or multiple-choice options). Excluded are moves that actually evaluate, select among, or confirm those options, as well as factual recall or computational steps. | "Given these considerations, let's think about what a multiple-choice question might look like and how the reasoning would apply to potential answers." |
| **Performing Calculations** | This cluster includes sentences that initiate and carry out concrete arithmetic, algebraic, or calculus operations – such as computing numerators or denominators, simplifying expressions, substituting values, multiplying, dividing, integrating, or factoring. They are typically signaled by imperative or invitational phrases like "Let's calculate...," "Now, let's perform the division," or "Simplify the expression," and they mark the direct execution of computational work. This category does not include higher-level planning of which operations to do next, statements of uncertainty, recollections of definitions, or hypothesis generation; it is strictly the step-by-step carrying out of calculations. | "Now, let's substitute these values into the equation." |
| **Concept Elaboration** | This category comprises sentences that supply explanatory content about ideas, phenomena, or terms. They often define or clarify what something is ("Firms are price takers, meaning..."), label and describe a concept using a "Term: explanation" structure ("Social Benefit: Better parenting can lead..."), or outline possible effects or functions using modals ("could," "might," "can"). These sentences do not propose hypotheses for later testing, express uncertainty about facts, plan future reasoning steps, or state assumptions; rather, they purely describe or elaborate existing concepts and their attributes. | "They may see technological innovation as a means to both mitigate environmental issues and maintain economic growth." |
| **Stating Given Information** | These sentences present the explicit details and parameters of a scenario – numerical values, contextual facts, and conditions – serving as the foundational premises for subsequent analysis. Included are declarative statements like "The mass of iron(III) oxide is 31.94 kg," "An 18-year-old man is brought to the emergency department," or "Pressure = atmospheric pressure," which simply record what is provided by the problem. Excluded are any sentences that outline reasoning steps, express uncertainty, propose hypotheses, or carry out calculations. | "The plate is 2 ft in length and initially at a surface temperature of 500°F. The ambient air temperature is 100°F, and the air flowrate is 150 ft/sec." |

*Continued from previous page*

| Category | Description | Top-Activating Example |
|---|---|---|
| **Case-by-Case Evaluation** | This cluster consists of sentences that systematically apply given conditions or values to derive immediate, concrete outcomes – such as evaluating a logical proposition under specific truth-value assignments, substituting a numerical value into an expression, or listing which cases satisfy a condition. These statements typically begin with "If," "When," "For," or enumerate outcomes ("There are 2 outcomes where w = 1: …"), and they produce explicit results (variable assignments, truth evaluations, covered edges, buffer-overflow conclusions, etc.). This category does not include speculative hypotheses, expressions of uncertainty, planning of future steps, or the recall of general definitions; instead, it focuses on the execution of rules in individual scenarios. | "- With probability $1/4$, we pick $W_3$ and then pick its neighbor at position 2, turning $W_2$ black: $B, B, W, B$." |
| **Problem Decomposition** | These sentences serve to organize the reasoning by dividing the overall task into smaller, ordered sub-steps and by restating or summarizing the problem at hand. They include explicit forward-looking markers and headings such as "Let's break down the problem step by step," "### Step 1: …," "### Problem Summary," and "To solve this problem, we need to…." They guide the structure of the solution without yet performing computations or applying specific domain knowledge. They do not include the actual execution of calculations, recall of definitions, statements of uncertainty, or the formulation of hypotheses. | "Let's break down the problem step by step." |
| **Recalling Mathematical Formulas** | These sentences function to retrieve and state known equations or definitions – such as physical laws, constitutive relations, or standard mathematical identities – as building-block premises for subsequent analysis. They include isolated formulaic expressions (e.g. Q = kA·ΔT/L, P = F/A, f(x)=$\sum a_n$cos(nx)) presented without immediate manipulation. They do not propose new hypotheses, outline next steps, express uncertainty, or perform calculations; their sole role is to supply canonical relationships from memory. | $Q = kA\frac{T_1 - T_2}{L}$ |

## G.9. QwQ-32B (Layer 27, Dict Size 10)

*Table 14.* Reasoning taxonomy for QwQ-32B (Layer 27, Dict Size 10). Categories are identified by our SAE; examples are top-activating sentences.

| Category | Description | Top-Activating Example |
|---|---|---|
| **Speculating answer choices** | These sentences generate plausible answer options or interpretations when the original question's list of choices is missing or unclear. They propose various potential answers or question contexts – using hedges and modals such as "maybe," "alternatively," "but perhaps," or "one possibility is" – to reconstruct what the expected options might have been. This category includes sentences that enumerate hypothetical answer forms or angles to fill gaps in the prompt. It does not include stating fixed assumptions, outlining next procedural steps, recalling formal definitions, or simply expressing generic uncertainty without proposing specific possibilities. | "Alternatively, maybe the question is from a test and the options were not given here, but the user wants the answer." |

*Continued from previous page*

| Category | Description | Top-Activating Example |
|---|---|---|
| **Proposing Additional Angles** | This cluster captures the speaker's act of brainstorming extra theories, frameworks, examples, or factors relevant to the current topic. Each sentence adds another possible perspective – often prefaced by "also," "another," "maybe," or "for example" – to broaden the conceptual space under consideration. These utterances do not commit to any option, plan a specific next step, or state formal assumptions; they simply enumerate potential ideas or angles to explore further. | "Another angle is the role of cultural influence in conflict and cooperation." |
| **Arithmetic Computation Steps** | These sentences carry out concrete numeric or algebraic operations – multiplying, dividing, adding, subtracting, and estimating – to advance the solution. They show step-by-step evaluations (e.g., "20.0004 = 0.0008," "9600.3353 $\approx$ 321.6," "4184*1.06 = 4435.04") often using approximations or unit conversions. This category includes any intermediary arithmetic or algebraic calculation explicitly stated in service of solving a problem. It does not include recalling formulas, stating assumptions, planning next steps, or expressing uncertainty. | "First, numerator: 18.288 0.3175 $\approx$ 18.288 0.3 = 5.4864, plus 18.288 * 0.0175 $\approx$ 0.320, so total $\approx$ 5.8064 cm$^2$/s. Divide by $\nu$: 5.8064 / 0.0089 $\approx$ 652.4." |
| **Exploring Conditional Outcomes** | These sentences systematically propose hypothetical "if–then" scenarios to uncover potential causal consequences or feedback effects of a change. They typically begin with markers such as "if," "when," "but if," "so," or "this might," and then articulate what could follow under those conditions. Included are speculations about how one variable or event might influence another, often hedged by modals ("might," "could," "may"). Not included are mere expressions of uncertainty, planning statements, or recalls of definitions; rather, these are focused on mapping out possible causal chains. | "So even if the government cuts taxes now, people might not spend the extra money because they expect to pay more taxes later, thus negating the intended stimulus." |
| **Recalling Definitions and Formulas** | These sentences serve to retrieve and state established domain knowledge – basic definitions, canonical formulas, standard properties, physical constants, chemical equations, and legal definitions – from memory as immediate premises for further reasoning. They often open with or include phrases like "I remember that...," "Recall that...," "That's because...," or directly recite well-known equations (e.g. the quadratic formula, distributive property, reaction equations). This category does not include proposing new hypotheses, planning next steps, expressing uncertainty about the overall approach, or laying out assumptions to be tested. | "That makes sense because division is the inverse operation of multiplication." |
| **Applying Legal Rules to Facts** | These sentences perform the core task of legal reasoning by stating specific legal principles (statutes, case holdings, doctrinal tests) and immediately mapping factual circumstances onto those principles to draw tentative conclusions. They often use "if...then..." constructions, cite controlling authorities (e.g., Supreme Court cases), invoke elements like duty, breach, causation, or mention defenses and exceptions. Included are tentative inferences about liability, admissibility, or enforceability (signaled by modals such as "may," "might," "could"). Not included are higher-level planning statements (e.g., "First I will..."), admissions of broad uncertainty without rule invocation (e.g., "I'm not sure"), or mere recall of definitions absent immediate fact-to-rule application. | "Under Steffel, the Court allowed such a suit when there's a credible threat of prosecution." |

*Continued from previous page*

| Category | Description | Top-Activating Example |
|---|---|---|
| **Recall of Standard Equations** | These sentences perform the cognitive operation of retrieving and stating a memorized formula or mathematical relationship to serve as a building block for further calculation. They typically begin with phrases like "The formula is," "Recall that," or simply present an equation (e.g., "Q = m·c·$\Delta$T"). Included are any declarations of domain-specific laws, definitions, or canonical equations without accompanying justification, derivation, or evaluation. Not included are sentences that plan next steps, propose hypotheses, acknowledge uncertainty, or establish working assumptions. | "Using Young's modulus formula, the strain ($\Delta$L / L) is equal to the stress ($\Delta$T / A) divided by Y. Wait, actually, the formula is:" |
| **Articulating the Next Reasoning Goal** | These sentences explicitly state the immediate question or sub-problem the reasoner intends to address, effectively framing the next step in the solution. They typically begin with formulations like "Okay, so I need to figure out...," "First, I need to recall...," or "The key here is to figure out...," thereby identifying what fact to retrieve or which option to evaluate. This category includes any utterance that focuses attention on what the reasoner must determine next. It does not include sentences that retrieve a definition, propose a hypothesis, express uncertainty about an answer, or lay out detailed multi-step methods beyond stating the goal. | "Okay, so I need to figure out which situation would most likely increase my aunt's demand for labor in her apple pie business." |
| **Meta-Cognitive Check-ins** | These sentences are self-directed prompts that signal the reasoner is pausing to reflect, reassess, or verify before proceeding. They include brief interjections or scaffolding phrases such as "Let me think," "Wait, let me confirm," "Hmm, let me check," and "Let me think again," which regulate the pace and monitor the internal reasoning process. They do not introduce new content, explicit hypotheses, or detailed next-step plans; rather, they serve purely as momentary cognitive pauses or self-checks. This category excludes substantive claims, mathematical definitions, or concrete planning instructions. | "Let me think." |
| **Answer Declaration** | These sentences explicitly state the result of a reasoning step or the final solution, often using discourse markers like "Therefore," "So," or "Hence" followed by a specific answer (e.g., a number, a choice, true/false). They serve to wrap up a calculation or logical deduction by announcing the conclusion reached. This category includes any phrasing that unambiguously asserts "the answer is..." or "the answer should be...," providing closure to the preceding work. It does not include speculative, planning, uncertainty, or definitional statements – only those that declare the outcome. | "Therefore, the answer should be 25%." |

