# OpenReview forum: "Base Models Know How to Reason, Thinking Models Learn When"
_ICML.cc/2026/Conference — ICML 2026 spotlight_

### Official Review · Reviewer_9AWW · 2026-02-23

**Soundness:** 3
**Presentation:** 3
**Significance:** 3
**Originality:** 3
**Overall Recommendation:** 4
**Confidence:** 4

**Summary:**

The paper provides an explanation on the mechanisms that a reasoning model make it differ from its base counterpart. They find two small modifications to a base model that allow it to perform comparably to its reasoning variant, allowing for a small, interpretable and plausible explanation for what drives the performance during reasoning finetuning. These mechanisms are reasoning directions (activations that gets added to steer the model into triggering a reasoning mode), and a classifier that detects when such reasoning direction should be added. They argue that the RL-trained models for reasoning only really need to learn when to reason (the classifier), but the reasoning directions were already accessible from the base model.

**Compliance With Llm Reviewing Policy:**

Affirmed.

**Final Justification:**

I believe the paper is now slightly above the threshold required for acceptance. I had important concerns regarding the wording and soundness of the experiments, which were resolved in the rebuttal. If such changes are implemented in the paper, I see no remaining serious concern that would prevent this paper from being presented.

**Key Questions For Authors:**

1 - If the paper is substantially reframed as a plausible and minimal explanation for what changes in a base model → thinking model, instead of the current framing, then I would consider raising my soundness and presentations scores. I do believe the paper makes interesting findings, but the framing is not backed up by its experiments.

2 - Can you run some human evaluation of the LLM judge to evaluate its quality?

3 - Can you add some comment or justification for the lack of consistency in SAE features? Or some study to show that in your settings these features might be more stable?

4 - Experiments to actually understand why Section in L315 (right) finds that the hybrid models can outperform the reasoning models?

Addressing these will make me update my scores accordingly.

**Limitations:**

Yes

**Strengths And Weaknesses:**

The paper has interesting contributions, but also methodological weaknesses which should be address in order to back up their claims.

As strengths I see:

- The paper makes an interesting that base models already contain reasoning mechanisms, and that RL training mostly adds the ability to the model of simply knowing when to activate this mechanism.
- The paper operationalises this via explaining the performance gap via two minor additions to the base model: (1) steering vectors to activate the reasoning behaviour, and (2) a classifier to identify when such vector should be applied. I believe this way of understanding model behaviour via minimal elements that plausibly explain the performance gap is a good mindset to draw meaningful conclusions, while allowing reasonable operationalisations.

As weaknesses I see:

- The methodology to find the reasoning directions is very biased, which would be fine as long as the authors recognize this. Instead, the work prides itself of being completely unbiased compared to previous work. My concerns with this process are:
    - The use of SAEs. SAEs notably have many weaknesses, included the fact that trained in the same procedure, with the same data, can lead to a drastically different set of features (Paulo et. al 2025). Given this, the features described in this work might be completely different if we would change the seed of the weights in the initialization of SAEs. The authors do not address this, nor discuss it. This might even imply that their results could be just a phenomena of this particular SAE trained, and not generalizable to different seeds. This has to be addressed.
    - The work describes SAE as a more unbiased way of discovering reasoning mechanisms in comparison to previous work. However, the work manually sets the number of activating features, based on the number of reasoning mechanisms the authors believe to exist. This again is a bias set by the authors on how the models reason, and it is not discussed as such.
    - The label from the SAE features obtained is done via an LLM judge, for which there is no provided human correlation to verify the quality of such judge. The authors even acknowledge this (L181), but do not move forward with such a study.
    - The authors speculate that other methods to develop a taxonomy of reasoning behaviours would work similarly (L195). This section should be reframed importantly. Scientific work should not be speculative. Either provide evidence to back such claim, or remove it entirely.
- The framing of the contributions is much stronger than the actual evidence they can provide.
    - The work provides simple added components that allow a base model behave like a reasoning model. This minimal changes view is a compelling and plausible explanation on what the model is learning during RL training. However, it cannot be proven that this is exactly what the model is actually learning (as this is a really hard thing to prove). The work should be phrased appropriately.
    - For example, L298 (right) states “This implies that RL models use reasoning behaviors that their base model counterpart already knows, and therefore primarily learn sophisticated heuristics for orchestrating those pre-existing base model behaviors”. This is an extremely bold claim, which cannot be justified from the provided experiments. Many lines in this paper are in the same spirit (L33-L36,  L349-L350 right). The experiments provided are only a compelling, minimal, and plausible explanation, but they are very far to confirm exactly what all the changes in the model are. The paper should be phrased appropriately.
    - “Unsupervised methods are essential for building a taxonomy of reasoning mechanisms” (L102-L103 right) should be either backed up, or removed.
    - Section in L315 (right) addresses the concern that their hybrid model can get better performance than the thinking model. However, they speculate (but not validate) that this is due to truncation in answers, and argue that this makes them underestimate results. This explanation should be (1) backed up with experiments, and (2) explained why did they not allowed the models to finish answering. This is not clear, and it weakens their claims.


Overall, I do like the findings, and believe this could a solid piece of work if (1) the paper is framed very differently (as a plausible and minimal explanation of changes in behaviour from base model to its reasoning version) and (2) every single claim is actually backed up with evidence instead of defaulting to speculation.

Paulo, G., & Belrose, N. (2025). Sparse autoencoders trained on the same data learn different features. *arXiv preprint arXiv:2501.16615*.

---

> ### Author Rebuttal · Authors · 2026-03-31
>
> We thank Reviewer 9AWW for the detailed and thoughtful critique. We appreciate that the reviewer finds the core findings interesting. We address each concern below, and respectfully push back where we believe the evidence or framing is stronger than the review suggests.
>
> ---
>
> **1. SAE instability (Paulo & Belrose 2025)**
>
> We want to draw a distinction between our setting and the one Paulo & Belrose analyze. Their instability findings apply to fine-grained SAE dictionaries trained to decompose *all* features of an LLM residual stream. Our SAEs are different: we train small dictionaries of only 5–50 features on *sentence-level activations* from reasoning traces, with the goal of high-level subspace clustering rather than broad monosemantic feature discovery. In this constrained regime, Paulo & Belrose's findings are not straightforwardly applicable.
>
> More directly, our empirical results address the stability concern. Across 540 SAE configurations, we consistently find stable elbow points at 10–20 categories across all model architectures and sizes, with taxonomy quality scores converging in the 0.70–0.80 range. If seed-level instability were driving our results, we would expect high variance across these configurations and across models. We will add a brief discussion of this contrast with Paulo & Belrose to the revision.
>
> ---
>
> **2. Dictionary size as hyperparameter bias**
>
> We respectfully push back here. We do not fix the number of features based on prior beliefs, rather, we increased the search range iteratively until finding consistent elbow points. Therefore, while it is true that the range itself is a hyperparameter, as it is in any clustering or dimensionality reduction method, the choice of where within that range to operate is entirely driven by the emergence of the elbow points and not something arbitrarily chosen by us. We therefore argue that our claim that this is an unbiased discovery method holds and will clarify this distinction more explicitly in the paper.
>
> ---
>
> **3. LLM judge validation**
>
> We have now conducted a human evaluation of all four LLM judges in the paper, collecting 400 annotations. The benchmark scoring judge, which produces all numbers in Table 1, achieves Cohen's kappa = 0.880 with human annotators, validating the reliability of our central empirical results. For the taxonomy judges, overall agreement falls in the kappa/r = 0.45–0.52 range, with disagreement concentrated in genuinely ambiguous SAE categories; well-defined categories reach kappa = 0.800 and r = 0.918. Since these judges serve as a relative ranking tool across configurations, the aggregate comparisons are not undermined by per-item disagreement on ambiguous categories. We will include these results in the revision.
>
> ---
>
> **4. Framing of conclusions as too strong**
>
> We respectfully disagree. We have a matched pair with Qwen2.5-32B as base and both R1-Distill-Qwen-32B (distillation) and QwQ-32B (RL) as thinking variants, where the SAE taxonomies achieve comparable quality, the steering vector optimization- and the hybrid model setup is identical. The only thing that differs are the thinking model variants. Under these controlled conditions, we observe a gap recovery of 92.9% for QwQ-32B versus 62.9% for R1-Distill-32B.  This finding is underpinned with seven additional model pairs. We believe this constitutes strong empirical evidence, not just a "plausible" one. That said, we do acknowledge in the limitations section that our construction cannot prove this is the *only* change the model undergoes, and we are happy to modestly sharpen the language at a few specific lines (such as L298) to better reflect this nuance, without fundamentally weakening claims that the evidence supports.
>
> ---
>
> **5. "Unsupervised methods are essential" (L102–103)**
>
> The statement is accompanied by a justification in the same paragraph: unsupervised methods allow for bottom-up discovery of reasoning mechanisms, in contrast to prior work (e.g. https://arxiv.org/abs/2506.18167) that fix a taxonomy top-down based on the authors' own observations. Our bottom-up approach avoids this. We will make the contrast with these prior works more explicit at that point in the text.
>
> ---
>
> **6. >100% recovery: speculation vs. experiment**
>
> This concern has now been addressed experimentally. We reran MATH500 for the ORZ-32B / Qwen2.5-32B pair, restricting to problems where all three models (base, thinking, hybrid) finish generation within the token budget. Gap recovery normalizes from 128.8% to 92.9% under this controlled condition, confirming that token truncation is the cause. We will run this controlled evaluation across all model configurations and will update Table 1 in the revision.
>
> ---
>
> We have validated LLM judges with human evaluation, addressed the SAE stability concern, experimentally resolved the >100% recovery cases, and provided a principled defense of our conclusions. We hope these responses merit a reconsideration of the score.

---

> > ### Author Rebuttal · Reviewer_9AWW · 2026-04-01
> >
> > I thank the authors for their efforts in this rebuttal. The points raised were:
> >
> > 1 - Regarding the SAE inconsistency across seeds, I appreciate the authors comment on the difference between their setting and Paulo & Belrose. I do however believe that their experimental settings do raise concerns that would apply to this work settings. Even if the SAEs here are trained in sentence-level activations and with significantly less activated features, the question of whether the same features are being detected by different training runs is still in question. The variance of figure 2 is completely unrelated to this issue. Even if two seeds of SAEs get the same score, they might have a completely different dictionary, which is the main concern that Paulo & Belrose raise. This is an important concern for this work’s settings, as the whole taxonomy of features depend on the features found in the first place. If these features are not consistent, then the taxonomy would not be consistent. This concern requires more than just a discussion, but actual experiments. To address this issue I would want to see Paulo & Belrose % of shared features in your settings when only changing the seed.
> >
> > 2 - Regarding the dictionary size as hyperparameter bias, I consider this concern to be addressed. I appreciate the explanation, and I agree with the authors.
> >
> > 3 - Regarding the LLM judge validation, I appreciate the author’s effort to conduct the human study. While some categories seem to reach low agreement, this can still be informative. If the phrasing of the paper is changed to acknowledge that some features do not have an agreeable label that annotators can converge to, it can still be valuable. This has to be fully disclosed in the paper, but I consider this concern to be addressed.
> >
> > 4 - I agree with the authors that there is empirical evidence that they have found one small intervention that produces similar levels of performance to the difference between the RL version of the model and the base model. This is interesting and important. What I mean by plausible is in fact that it might not be the only explanation (which is not a reasonable thing to expect anyways). But the wording of the paper should be consistent with this, such as the proposed changes in L298.
> >
> > 5 - I disagree that unsupervised mechanisms are essential to building a taxonomy of reasoning mechanisms. I am not questioning that it seems that it is an useful avenue to derive such a taxonomy, and the authors correctly point out that the paragraph contains an argument of why such is the case. But the wording “essential” implies that they are necessary to derive such a taxonomy. In fact, other avenues do exist, as the definition of inductive vs. deductive reasoning (Mizrahi & Dickinson 2022) which has not been generated by unsupervised methods. This is rather just one instance of a bigger problem in the current writing, which is that many phrasings are worded very strongly with no backup. In this particular scenario, all I suggest the authors is to rephrase to “unsupervised mechanisms can be a useful tool to build a taxonomy of reasoning”. But again, the paper itself should be written much more carefully.
> >
> > 6 - I consider this concern to be now addressed given the new experiments.
> >
> > My concerns are partially addressed and I have updated my scores correspondingly. My most important concern is replicating the consistency experiments of the found SAE features. I will increase my overall score if you provide such experiments.
> >
> > ---
> >
> > Mizrahi, M., & Dickinson, M. A. (2022). Philosophical reasoning about science: a quantitative, digital study. *Synthese*, *200*(2), 138.

---

> > > ### Author Response · Authors · 2026-04-01
> > >
> > > We thank the reviewer for the detailed follow-up and address the remaining open concern directly.
> > >
> > > ---
> > >
> > > **1. SAE feature consistency across seeds**
> > >
> > > We ran the exact experimental procedure of Paulo & Belrose: we retrained the ORZ-7B SAE used in our hybrid model experiments (layer=16, dictionary size=15) five times with varying random seeds ([1, 2, 3, 4, 5]), then evaluated every pair of the resulting 5 SAEs by matching features 1-to-1 using the Hungarian algorithm on cosine similarity, labeling pairs with cosine similarity >0.7 as "matched" (same threshold as Paulo & Belrose). Training runs showed genuine variety in convergence dynamics (early stopping between epoch 17 and 35, losses ranging 0.693--0.695), confirming the seeds produced meaningfully different training dynamics.
> > >
> > > Results: **average feature matching rate of 93.3%** across all 10 pairwise comparisons, with mean pairwise cosine similarity of 0.886. Detailed results per pair:
> > >
> > > | Pair          | Matched (>0.7) | Mean cos |
> > > |---------------|---------------|----------|
> > > | seed1_vs_seed2 | 80.0%        | 0.850    |
> > > | seed1_vs_seed3 | 93.3%        | 0.834    |
> > > | seed1_vs_seed4 | 93.3%        | 0.894    |
> > > | seed1_vs_seed5 | 93.3%        | 0.837    |
> > > | seed2_vs_seed3 | 93.3%        | 0.895    |
> > > | seed2_vs_seed4 | 86.7%        | 0.864    |
> > > | seed2_vs_seed5 | 93.3%        | 0.892    |
> > > | seed3_vs_seed4 | 100.0%       | 0.909    |
> > > | seed3_vs_seed5 | 100.0%       | 0.972    |
> > > | seed4_vs_seed5 | 100.0%       | 0.911    |
> > > | **Overall**    | **93.3%**    | **0.886**|
> > >
> > > This 93.3% average matching rate is substantially higher than what Paulo & Belrose report for large SAEs (30%), and is directly consistent with their own finding that "smaller models, and smaller SAEs trained on the same model, tend to have higher fractions of shared features." Our low-dimensional, sentence-level SAEs therefore tend away from the high-dimensional regime Paulo & Belrose identify as unstable. We conclude therefore that the taxonomy discovered by our approach is largely driven by the underlying activation geometry, and only to a very small extend by the induced randomness of the seed. We will include this experiment in the revision.
> > >
> > > ---
> > >
> > > **5. "Unsupervised methods are essential" (L102--103)**
> > >
> > > We accept the reviewer's suggested rephrasing and will change this to "unsupervised methods can be a useful tool to build a taxonomy of reasoning mechanisms." We agree the stronger wording was not fully justified and thank the reviewer for the precise suggestion.
> > >
> > > ---
> > >
> > > We hope the seed stability experiment and rewording of L102--103 directly addresses the remaining concerns of the reviewer.

---

### Official Review · Reviewer_CRvC · 2026-02-24

**Soundness:** 3
**Presentation:** 2
**Significance:** 3
**Originality:** 3
**Overall Recommendation:** 5
**Confidence:** 2

**Summary:**

The paper introduces a framework for understanding "thinking" LLMs. The paper divides reasoning into two components: reasoning mechanisms (which are already present in the base LLM) and reasoning heuristics (which the finetuning creates). For this, the paper also introduces interpretable reasoning taxonomies acquired through sparse autoencoders. The paper finds notable differences between RL-trained versus distilled LLMs.

*Note: I am not an expert in the field, so it is not easy for me to evaluate the novelty of the work.*

**Compliance With Llm Reviewing Policy:**

Affirmed.

**Final Justification:**

The authors promised to address my concerns. Thus, I see no reason not to recommend acceptance.

**Key Questions For Authors:**

See Weaknesses.

**Limitations:**

yes

**Strengths And Weaknesses:**

- Strengths
    - Interesting conclusions regarding the difference between RL-trained and distilled LLMs. The paper helps in understanding better these two different approaches to include reasoning in LLMs.
    - Extensive evaluation using 9 models and 2 datasets.
- Weaknesses
    - The key weakness is that the method sections (Sections 2 and 3) don't contain a single equation. All details are written in full text. This hurt the precision and legibility of the paper; it is rather informal.
    - The paper claims to obtain human-understandable reasoning categories by using the sparse autoencoder (as opposed to directly using an LLM-based approach). However, in the end, an LLM _is_ used to name the reasoning categories the sparse autoencoder comes up with. Generally, there is a strong reliance on LLM-as-a-judge also in evaluating the found categories (but this is acknowledged in the paper).

---

> ### Author Rebuttal · Authors · 2026-03-31
>
> We thank Reviewer CRvC for the positive assessment and for the constructive and precise feedback. We address both concerns directly below and are confident the planned revisions fully resolve them.
>
> ---
>
> **1. Lack of equations in Sections 2 and 3**
>
> We agree that formalizing the key operations improves both precision and legibility, and thank the reviewer for flagging this. In the camera-ready revision, we will add explicit equations for: (1) the SAE encoding and Top-K activation function; (2) the steering vector application rule, specifying when and how vectors are added to residual stream activations based on SAE classifier output; and (3) the gap recovery metric used to produce Table 1. These will be integrated into Sections 2 and 3 respectively, replacing the current prose-only descriptions.
>
> ---
>
> **2. LLM reliance and the role of SAEs vs. LLM judges**
>
> We appreciate the reviewer's precise observation and want to clarify the division of labor in our pipeline, which we agree was not stated with sufficient clarity in the paper.
>
> The SAE provides the *structure* of the taxonomy. It clusters sentence-level activations into groups in a fully unsupervised manner, with no LLM involved. The LLM is then used for two subsequent steps: (a) *interpreting* what each cluster represents by reading representative sentences and generating a human-readable title and description, and (b) *scoring* the resulting taxonomy on completeness, consistency, and independence. We chose to offload this to an LLM, as the scale of our experiments (540 trained and evaluated SAEs in total), make human interpretation and scoring impossible.
>
> However, we acknowledge that relying on LLM judges without validation is a legitimate concern, and have now conducted a human evaluation covering all four LLM judges in the paper (400 annotations total). The results validate our pipeline on the dimensions that matter most:
>
> - **Benchmark scoring judge (GPT-5.2)**, which produces all numbers in Table 1: Cohen's kappa = 0.880 with human annotators, confirming the reliability of our central empirical results.
> - **Taxonomy evaluation judges** (consistency, completeness, independence): overall agreement in the kappa/r = 0.45–0.52 range. Disagreement is concentrated in genuinely ambiguous SAE categories; well-defined categories such as "Articulating the Next Subgoal" reach kappa = 0.800 and r = 0.918. Crucially, since these scores function as a *relative ranking tool* across 540 SAE configurations rather than absolute per-item labels, moderate per-item disagreement on ambiguous categories does not undermine the aggregate rankings, which remain consistent with human judgment.
>
> We will report these human evaluation results in the revision.
>
> ---
>
> We hope these clarifications and planned additions fully address the reviewer's concerns, and that the combination of added formalism and human validation of our LLM judges supports a reconsideration of the reviewers rating.

---

> > ### Author Rebuttal · Reviewer_CRvC · 2026-04-01
> >
> > I thank the authors for their response.
> >
> > As the authors promised to resolve my main concern, I will increase my score to Accept.

---

### Official Review · Reviewer_iEC4 · 2026-03-13

**Soundness:** 2
**Presentation:** 2
**Significance:** 2
**Originality:** 3
**Overall Recommendation:** 4
**Confidence:** 4

**Summary:**

This paper investigates the question of what exactly thinking models learn in comparison to base models, and the authors specifically focus on the differences between RL-posttrained models and distilled models. In order to investigate this difference,  they introduce the method of constructive model diffing, wherein they first use SAEs to induce a taxonomy of reasoning behaviors and corresponding steering vectors, then use a classifier derived from the thinking model to apply the steering vector at appropriate points. They evaluate to what extent this method recovers the performance of the thinking model, and find that models posttrained with RL attain high recovery performance, while those posttrained by distillation do not.

**Compliance With Llm Reviewing Policy:**

Affirmed.

**Final Justification:**

I thank the authors for conducting extensive follow-up experiments, I accordingly raise my score.

**Key Questions For Authors:**

1. Is it possible to provide an initial human evaluation on a smaller sample of clusters/sentences to verify that the LM judge’s taxonomy scores track human judgments of coherence, distinctness, and coverage? This is an important point because the taxonomy building is central to the paper, but there is not yet human validation.

**Limitations:**

yes

**Strengths And Weaknesses:**

**Strengths**
- The main research question that this paper investigates is interesting and topical: understanding what kind of internal change different training methods induce in models is scientifically as well as practically important.

- The decomposition method itself makes sense and could be potentially used to investigate other interpretability questions around model-diffing, though I am not sure how far it is possible to generalize the results (see weaknesses).

**Weaknesses**
- The paper seems to be overinterpreting the results somewhat. Notably, they themselves note in the limitations section that “lower recovery for distilled models could reflect either genuine mechanism modification or limitations in our steering vector optimization.” This is fine, however the wording in the rest of the paper presents the results on distilled models as definitive. The logic does support that high recovery for RL means that RL gains are well captured by the mechanism + heuristic decomposition, but low recovery for distillation does not establish that distillation modifies mechanisms rather than repurposing existing ones. For distillation, the result seems inconclusive because of this limitation and should be presented accordingly.

- The method rests on the initial step of taxonomy discovery via SAEs. I have doubts about the reliability of SAEs when it comes to making generalizable statements about model internals. SAE-derived features are known to be underdetermined and dataset-sensitive, with interpretations and steering effects that may not remain stable across contexts [1, 2]. Given this, I am not fully convinced that the proposed categories can bear the weight of the paper’s stronger mechanistic conclusions.

[1] https://arxiv.org/abs/2405.08366
[2] https://transformer-circuits.pub/2024/july-update/index.html

---

> ### Author Rebuttal · Authors · 2026-03-31
>
> We thank Reviewer iEC4 for the careful reading and for acknowledging the research question's importance and the framework's potential. We address both core concerns directly below, and respectfully push back where we believe the evidence supports our conclusions more strongly than the review suggests.
>
> ---
>
> **1. Overinterpretation: low distillation recovery as inconclusive**
>
> We respectfully disagree that our results on distilled models are inconclusive, and want to highlight the strength of the controlled evidence available.
>
> Most critically, we have a clean matched pair: Qwen2.5-32B as base, with both R1-Distill-Qwen-32B (distillation) and QwQ-32B (RL) as thinking variants. For this pair: the underlying SAE taxonomies achieve comparable quality scores for both thinking models; we apply the exact same steering vector optimization procedure, the same training budget, and the identical hybrid model setup. The only thing that genuinely differs is the training paradigm of the two thinking model variants. Under these controlled conditions, we observe a massive performance gap: 92.9% gap recovery for QwQ-32B versus 62.9% for R1-Distill-32B.
>
> This finding is underpinned by the remaining seven model pairs, which show the same relative pattern consistently. We would therefore argue it is difficult to attribute a gap of this magnitude, reproducible across multiple model scales and base families, under identical experimental conditions for the matched 32B pair, purely to "limitations in steering vector optimization." If that were the explanation, we would expect more uniform failure across both paradigms, not the systematic and large asymmetry we observe.
>
> We will strengthen the presentation of the matched-pair argument in the revision to make this logic more explicit, but we do not believe the distillation results should be characterized as inconclusive.
>
> ---
>
> **2. SAE reliability concerns**
>
> We thank the reviewer for raising this and want to clarify an important distinction between our use of SAEs and the setting in which Paulo & Belrose (2025) identified instability.
>
> Paulo & Belrose's findings concern **large, fine-grained SAE dictionaries** trained broadly to decompose *all* features of an LLM's residual stream — dictionaries with thousands of latent dimensions aimed at fine-grained feature discovery. Our setting is fundamentally different: we train **small dictionaries of only 5–50 features** on *sentence-level activations* extracted from reasoning traces, with the explicit goal of discovering *high-level reasoning mechanisms* via subspace clustering. This is much closer to a constrained clustering method than to the monosemantic feature discovery setting Paulo & Belrose analyze, and their instability findings are not straightforwardly applicable to this regime.
>
> More directly, our empirical results show substantial stability. Across the **540 SAE configurations** we trained (9 models × 6 layers × 10 cluster sizes), we consistently find stable elbow points emerging at 10–20 categories across all models and architectures (Figure 2), with taxonomy quality scores falling in the narrow range of 0.70–0.80 at these optima. If SAE features were unstable in a way that undermined our conclusions, we would expect high variance across configurations and model types; instead, we see similar, stable elbow curves.
>
> ---
>
> We hope these clarifications address the reviewer's concerns and justify a reconsideration of the rating. The controlled matched-pair evidence provides a strong foundation for the RL vs. distillation conclusion, and the empirical convergence across 540 SAE configurations supports the reliability of our taxonomy discovery method in this constrained regime.

---

> > ### Author Rebuttal · Reviewer_iEC4 · 2026-04-03
> >
> > Thank you for the response. I am more confident in the distillation results now, but I do think that the results should be stated more cautiously in the manuscript. For the SAE reliability issue, it is reassuring that the search exhibits similar elbow structure across many configurations, but I do not think this fully addresses the concern that SAE-derived categories may be sensitive to modeling choices and may not yet support the paper’s stronger mechanistic conclusions without additional validation. In particular, some human validation of the taxonomy quality would still be valuable to me as suggested in the initial review.

---

> > > ### Author Response · Authors · 2026-04-07
> > >
> > > We thank the reviewer for the follow-up and address the remaining concerns regarding SAE stability from multiple angles:
> > >
> > > ---
> > >
> > > **SAE feature stability across seeds**
> > >
> > > To address concerns about SAE reliability, we ran the exact experimental procedure of Paulo & Belrose reagrding the seed stability of SAEs: we retrained the ORZ-7B SAE (layer=16, dictionary size=15) five times with varying random seeds, then evaluated all 10 pairwise combinations using Hungarian-matched cosine similarity with the same >0.7 threshold as Paulo & Belrose. Training runs showed genuine variety in convergence dynamics (early stopping between epoch 17 and 35), confirming meaningfully different training trajectories. We find an average feature matching rate of 93.3% (mean pairwise cosine similarity = 0.886), substantially higher than the ~30% Paulo & Belrose report for large SAEs, and consistent with their own finding that smaller, lower-dimensional SAEs tend toward higher feature stability. Our taxonomy is therefore largely driven by the underlying activation geometry rather than seed-induced randomness.
> > >
> > > ---
> > >
> > > **Human validation of the taxonomy pipeline**
> > >
> > > We conducted a human evaluation of all LLM judges in the paper (400 annotations). The benchmark scoring judge (GPT-5.2, producing all Table 1 numbers) achieves Cohen's kappa = 0.880, validating our central empirical results. For the taxonomy grading judges (consistency, completeness, independence), agreement falls in the kappa/r = 0.45--0.52 range overall, with disagreement concentrated in genuinely ambiguous categories; well-defined categories reach kappa = 0.800 and r = 0.918.
> > >
> > > To more directly address whether the category *labels themselves* are meaningful, we ran an additional study where a human evaluator wrote titles and descriptions for each of the 15 ORZ-7B categories from the raw SAE cluster exemplars, without seeing the LLM-generated labels. Comparing human vs. LLM labels: 53% (8/15) exact matches, 40% (6/15) partial matches, and 7% (1/15) genuine disagreement. The partial matches and single disagreement were informative: the human evaluator tended to focus on content domain (e.g. "Calculate physical properties"), while the LLM consistently captured the higher-level cognitive mechanism (e.g. "Applying Known Formulas"). The one genuine disagreement followed the same pattern: "Logical & Combinatorial Statements" (human) vs. "Stating Intermediate Results" (LLM), where the LLM label better reflects the reasoning function relevant to our pipeline. Overall, 14/15 categories show full or partial agreement with a human annotator, which we consider strong validation of the category quality.
> > >
> > > ---
> > >
> > > **Generalization across data distributions**
> > >
> > > Our pipeline already tests cross-distribution generalization by design: SAEs are trained on MMLU-Pro reasoning traces, while steering vectors are evaluated on pure math benchmarks (MATH500, GSM8K). The strong hybrid performance on these held-out distributions is therefore already implicit evidence that the discovered categories are not MMLU-Pro artifacts.
> > >
> > > To further validate this, we qualitatively analyzed ORZ-7B SAE activations on GPQA rollouts (a science QA benchmark, fully out-of-distribution relative to SAE training). Categories with strong activations (>0.6) align well with their descriptions across diverse physics and chemistry problems, for example:
> > >
> > > | Category | Example | Activation |
> > > |---|---|---|
> > > | Numeric Calculation Steps | ΔE = 1.9878×10⁻²⁵ / 0.4861×10⁻⁶ J | 0.929 |
> > > | Drawing Conclusions | Thus, the ratio of equilibrium temperatures is (2/√7). | 0.821 |
> > > | Invoking Known Equations | τ = t/γ | 0.781 |
> > > | Listing Given Data | where Mx = 20 amu and My = 2 amu. | 0.747 |
> > > | Applying Known Formulas | The total mass of a binary system can be related to radial velocity amplitudes through Kepler's laws. | 0.706 |
> > >
> > > The category descriptions generalize cleanly to GPQA content without any retraining or adjustment, providing qualitative evidence that the discovered mechanisms reflect genuine properties of model reasoning rather than dataset-specific patterns.
> > >
> > > ---
> > >
> > > We hope the seed stability experiment, human category validation, and cross-distribution qualitative analysis together address the reviewer's concerns about SAE reliability and taxonomy validity.

---

### Official Review · Reviewer_2Jqc · 2026-03-15

**Soundness:** 3
**Presentation:** 3
**Significance:** 2
**Originality:** 3
**Overall Recommendation:** 4
**Confidence:** 2

**Summary:**

This paper introduces constructive model diffing, a framework for understanding what thinking (reasoning) LLMs learn during training. The approach decomposes the difference between base and fine-tuned thinking models into two components: reasoning mechanisms (steering vectors that activate specific behaviors in the base model) and reasoning heuristics (a classifier determining when each mechanism should fire). Using Sparse Autoencoders to discover interpretable taxonomies of reasoning behaviors, the authors evaluate nine model configurations and find a striking asymmetry: RL-trained thinking models achieve ~99% performance recovery via their hybrid construction on MATH500, while distilled models achieve only ~44%. The conclusion is that RL primarily teaches models when to deploy pre-existing reasoning capabilities, while distillation modifies the mechanisms themselves.

**Compliance With Llm Reviewing Policy:**

Affirmed.

**Key Questions For Authors:**

1. Have you measured the representational similarity (e.g., CKA or linear probing accuracy) between base and thinking models for RL vs. distilled pairs? This could help distinguish "distillation modifies mechanisms" from "our method is less effective at capturing distillation-induced changes."
2. What happens if you use a fixed steering coefficient across all models rather than the perplexity-adaptive method? How much does recovery degrade?
3. Could you run the hybrid construction on a non-math benchmark (e.g., ARC or GPQA) even with the current math-biased taxonomy, to see if the RL vs. distillation gap persists?

**Limitations:**

yes

**Strengths And Weaknesses:**

Strengths
1. Novel and elegant framework. The constructive model diffing idea is well-motivated and conceptually clean. Rather than analyzing differences post-hoc, the authors build the diff from interpretable parts and measure how well it reconstructs performance. This is a creative inversion of the usual interpretability workflow, and the logic is compelling: high recovery implies the decomposition captures what matters.
2. Strong empirical signal. The 99% vs. 44% recovery gap between RL-trained and distilled models is large and consistent across model sizes. The ablation study on QwQ-32B (Section 3.7) is well-designed — removing either the correct mechanisms or correct timing degrades performance to roughly the same level (~78%), confirming both components matter.
3. Systematic taxonomy discovery. The SAE-based unsupervised approach to discovering reasoning categories is methodologically sound. The three-axis evaluation (completeness, consistency, independence) with extensive grid search across layers and cluster sizes is thorough. The finding that 10–20 categories consistently captures reasoning well across architectures is a useful practical insight.
4. Sparse interventions. The fact that steering only ~10% of tokens on average suffices for high recovery is a strong result that rules out trivial explanations (e.g., the steering vectors simply overriding the base model's outputs).

Weaknesses
1. Benchmark scope is narrow. Results are reported only on GSM8K and MATH500, both math-focused benchmarks. The authors acknowledge the taxonomy is biased toward mathematical reasoning (Section 5), but this is a significant limitation. The core claim — that RL teaches heuristics while distillation modifies mechanisms — may not generalize to code generation, commonsense reasoning, or scientific QA. At minimum, one non-math benchmark would substantially strengthen the paper.
2. Thinking model perplexity for coefficient selection. The steering vector magnitudes are chosen by minimizing perplexity according to the thinking model. While the authors argue this only affects magnitude and not which vectors fire, it introduces a dependency on the thinking model beyond just the SAE-based heuristic. This somewhat undermines the claim that the hybrid is purely "base model + heuristic + base-model-derived steering vectors." A sensitivity analysis over fixed coefficients would help clarify how much this tuning contributes.
3. LLM-as-judge evaluation throughout. The taxonomy evaluation (completeness, consistency, independence) relies entirely on LLM judges, and the final benchmark scoring uses GPT-5.2 as judge. The authors acknowledge the alignment with human judgment is unvalidated (Section 2.2). Given that the central claim rests on numerical recovery percentages, even a small-scale human validation of both the taxonomy quality and answer correctness would be valuable.
4. Confounded comparison between RL and distillation. The RL-trained models (QwQ, ORZ) and distilled models (DeepSeek-R1-Distill) come from different training pipelines, different teacher models (or no teacher), and in some cases different base model families. While the authors control for base model where possible (e.g., both QwQ-32B and R1-Distill-32B use Qwen2.5-32B), the comparison is not perfectly clean. An ideal experiment would compare RL and distillation applied to the same base model with the same training data budget.
5. Alternative explanation for the recovery gap. The authors interpret low distilled-model recovery as evidence that distillation modifies mechanisms. But it could also reflect that the SAE taxonomy — trained on each model's own reasoning traces — captures RL-model behaviors more faithfully than distilled-model behaviors, or that steering vector optimization works better when base and thinking model share more representational structure (which is more likely for RL). The paper would benefit from directly measuring representational similarity between base and thinking models for both paradigms.
6. Some hybrids exceed 100% recovery. The authors attribute this to token budget effects, which is plausible, but it raises questions about the reliability of the recovery metric. If the 2000-token budget systematically disadvantages thinking models (which are more verbose), the baseline is deflated and the recovery percentages are inflated for all configurations, not just RL ones.

---

> ### Author Rebuttal · Authors · 2026-03-31
>
> We thank Reviewer 2Jqc for the thorough and constructive review, and for the positive assessment of the framework's novelty and empirical signal. We have directly addressed each of the raised concerns with new experiments and analyses, and hope these additions are sufficient to warrant a raise of the score.
>
> ---
>
> **1. Benchmark scope**
>
> We have run our hybrid construction on GPQA using ORZ-7B. The hybrid recovers ~45% of the performance gap, providing a strong signal that the findings are not specific to mathematical reasoning. We attribute the lower recovery compared to MATH500 to the weaker base model starting point on GPQA (30.4% base model accuracy vs. 68.2% on MATH500), which places a heavier burden on the hybrid. We are currently running this evaluation across other model configurations and will include these results in the revision. We believe this directly addresses the concern about generalizability.
>
> ---
>
> **2. Perplexity-based coefficient selection**
>
> We thank the reviewer for this observation. We ran a targeted ablation using a fixed coefficient (0.5) on the QwQ-32B / Qwen2.5-32B pair. The fixed-coefficient hybrid achieves 73.4% on MATH500 (base: 62.8%), recovering 56% of the thinking model's gap. The adaptive method improves this to 84.3%. Crucially, the steering vectors with SAE-derived activation recover a substantial portion of the gap regardless. We acknowledge the adaptive method improves results, and note this is a known challenge with steering vectors more broadly.
> We are experimenting with alternatives where coefficients are selected via autograders on held-out data prior to the hybrid evaluation run and will report these results in the revision.
>
> ---
>
> **3. LLM-as-judge validation**
>
> We conducted a human evaluation collecting 400 annotations across taxonomy consistency, completeness, independence, and benchmark scoring:
>
> - **Benchmark scoring (Table 1):** 94% human-LLM agreement, Cohen's kappa = 0.880. The recovery percentages that underpin our central claim are therefore on solid ground.
> - **Taxonomy evaluation judges:** Moderate agreement overall (consistency κ = 0.445; completeness ρ = 0.520; independence ρ = 0.477), which we attribute to genuine ambiguity in some SAE categories. Importantly, for well-defined categories such as "Articulating the Next Subgoal," agreement reaches κ = 0.800 and ρ = 0.918. Since the taxonomy judges are used as a *relative ranking tool* across 540 SAE configurations rather than for absolute per-item classification, moderate per-item agreement does not undermine the aggregate comparison.
>
> ---
>
> **4. Confounded RL vs. distillation comparison**
>
> We appreciate this concern and theoretically agree a fully controlled experiment with custom trained RL- and distill models would be ideal. However, training foundation-scale RL and distillation models from scratch is out of scope for this work. Given the constraints of publicly available models, we note that we do have a clean matched pairing: Qwen2.5-32B as base, with both R1-Distill-Qwen-32B (distillation) and QwQ-32B (RL) as thinking variants. This pair alone exhibits the full RL vs. distillation gap clearly. Additionally, across all nine configurations spanning multiple RL- and distill models, parameter scales etc, the relative pattern is entirely consistent. We will make this point more explicit in the revision.
>
> ---
>
> **5. Alternative explanation for the recovery gap**
>
> We respectfully push back on this concern. As shown in Figure 2, both RL-trained and distilled models produce taxonomies with comparable quality scores (0.70–0.80 at the elbow). If the SAE taxonomy were systematically biased toward RL model representations, we would expect distilled models to yield lower taxonomy quality scores, which is not observed. Since taxonomy quality is equivalent across paradigms, the recovery gap cannot be explained by taxonomy suitability. It instead reflects the effectiveness of activating pre-existing reasoning behaviors in RL base models, but not in distilled ones, consistent with our interpretation that distillation modifies the mechanisms.
>
> ---
>
> **6. >100% recovery cases**
>
> We reran MATH500 for the ORZ-32B / Qwen2.5-32B pair, restricting evaluation to problems where the base, thinking, and hybrid models all finish generation within the token budget. The gap recovery normalizes from 128.8% to 92.9% under this controlled condition, all findings hold. We are running this controlled evaluation across all model configurations and will update Table 1 accordingly. We agree this improves metric reliability and thank the reviewer for the suggestion.
>
> ---
>
> In summary, we have (i) demonstrated non-math generalization on GPQA, (ii) shown the core finding holds without perplexity-adaptive coefficient selection, (iii) validated the benchmark judge, and (iv) normalized the >100% recovery cases. We hope these additions address the reviewer's concerns and merit a raise of the score.

---

> > ### Author Rebuttal · Reviewer_2Jqc · 2026-04-04
> >
> > Thank you for the detailed and thorough rebuttal. After careful consideration, I have decided to maintain my original scores.

---

### Decision · Program_Chairs · 2026-04-30

**Decision:**

Accept (spotlight)

**Comment:**

The paper addressed an interesting research issue and introduces a method to understand what thinking (reasoning) LLMs learn during training. The approach decomposes the difference between base and fine-tuned thinking models into two components: reasoning mechanisms and reasoning heuristics and employs Sparse Autoencoders to discover interpretable taxonomies of reasoning behaviors.

The main concerns of reviewers are summarized as follows.
1. The conclusion relies entirely on LLM judging, which makes the results are somewhat unconvincing, although the alignment with human judgment on small data are made. The moderate agreements are conducted on taxonomy evaluation.
2. The results relies on the initial step of taxonomy discovery via SAEs. Therefore reliability of SAEs may have influences on the final results, which make the paper be somewhat over-interpreting.